# The effect of intravenous hyoscine butylbromide on slow progress in labor (BUSCLAB): A double-blind randomized placebo-controlled trial

**Lise Christine Gaudernack**[1,2‡]*, **Angeline Elisabeth Styve Einarsen**[1,3‡], **Ingvil Krarup Sørbye**[1,3], **Mirjam Lukasse**[2,4], **Nina Gunnes**[5], **Trond Melbye Michelsen**[1,3]*

1 Department of Obstetrics, Division of Obstetrics and Gynecology, Oslo University Hospital, Oslo, Norway, 2 Department of Nursing and Health Promotion, Faculty of Health Sciences, Oslo Metropolitan University, Oslo, Norway, 3 Institute of Clinical Medicine, Faculty of Medicine, University of Oslo, Oslo, Norway, 4 Department of Nursing and Health Sciences, Faculty of Health and Social Sciences, University of South-Eastern Norway, Campus Vestfold, Borre, Norway, 5 Norwegian Research Centre for Women's Health, Oslo University Hospital, Oslo, Norway

‡ These authors share first authorship on this work.
* lisegaudernack@yahoo.no (LCG); trmi1@ous-hf.no (TMM)

**Data Availability Statement:** There are ethical and legal restrictions on sharing these clinical data. The primary outcome was time from administration of

## Abstract

### Background

Prolonged labor is a common condition associated with maternal and perinatal complications. The standard treatment with oxytocin for augmentation of labor increases the risk of adverse outcomes. Hyoscine butylbromide is a spasmolytic drug with few side effects shown to shorten labor when used in a general population of laboring women. However, research on its effect on preventing prolonged labor is lacking. We aimed to assess the effect of hyoscine butylbromide on the duration of labor in nulliparous women showing early signs of slow labor.

### Methods and findings

In this double-blind randomized placebo-controlled trial, we included 249 nulliparous women at term with 1 fetus in cephalic presentation and spontaneous start of labor, showing early signs of prolonged labor by crossing the alert line of the World Health Organization (WHO) partograph. The trial was conducted at Oslo University Hospital in Norway from May 2019 to December 2021. One hundred and twenty-five participants were randomized to receive 1 ml hyoscine butylbromide (Buscopan) (20 mg/ml), while 124 received 1 ml sodium chloride intravenously. Randomization was computer-generated, with allocation concealment by opaque sequentially numbered sealed envelopes. The primary outcome was duration of labor from administration of the investigational medicinal product (IMP) to vaginal delivery, which was analyzed by Weibull regression to estimate the cause-specific hazard ratio (HR) of vaginal delivery between the 2 treatment groups, with associated 95% confidence interval (CI). A wide range of secondary maternal and perinatal outcomes were also evaluated.

the Investigational Medicinal Product to delivery. The data underlying this outcome contains date and time of birth and cannot be shared. Moreover, combinations of the variables collected could possibly identify participants and thereby compromise participant privacy. In addition, the participants have not consented to sharing of collected data with international researchers according to GDPR recommendations. Contact email to the study sponsor: mirnyb@ous-hf.no.

**Funding:** Foundation Dam/ The Norwegian Women's Public Health Association has funded the PhD project and paid salary for AESE, MD. Grant number (2020/FO283405). Oslo University Hospital has founded the PhD project and paid salary for LCG, RM, MPH, PhD No Grant number available. The funding sources did not participate in the study's design, manuscript preparation, analysis and approval, or the decision to submit the manuscript for publication.

**Competing interests:** The authors have declared that no competing interests exist.

**Abbreviations:** BMI, body mass index; CI, confidence interval; CTG, cardiotocography; eCRF, electronic case report form; FA, full analysis; HR, hazard ratio; IMP, investigational medicinal product; ITT, intention-to-treat; IU, international unit; NICU, neonatal intensive care unit; OR, odds ratio; PI, principal investigator; PP, per-protocol; RCT, randomized controlled trial; SD, standard deviation; SHR, sub-hazard ratio; WHO, World Health Organization.

Time-to-event outcomes were analyzed by Weibull regression, whereas continuous and dichotomous outcomes were analyzed by median regression and logistic regression, respectively. All main analyses were based on the modified intention-to-treat (ITT) set of eligible women with signed informed consent receiving either of the 2 treatments. The follow-up period lasted during the postpartum hospital stay. All personnel, participants, and researchers were blinded to the treatment allocation. Median (mean) labor duration from IMP administration to vaginal delivery was 401 (440.8) min in the hyoscine butylbromide group versus 432.5 (453.6) min in the placebo group. We found no statistically significant association between IMP and duration of labor from IMP administration to vaginal delivery: cause-specific HR of 1.00 (95% CI [0.77, 1.29]; $p = 0.993$). Among 255 randomized women having received 1 dose of IMP, 169 women (66.3%) reported a mild adverse event: 75.2% in the hyoscine butylbromide group and 57.1% in the placebo group (Pearson's chi-square test: $p = 0.002$).

More than half of eligible women were not included in the study because they did not wish to participate or were not included upon admission. The participants might have represented a selected group of women reducing the external validity of the study.

## Conclusions

One intravenous dose of 20 mg hyoscine butylbromide was not found to be superior to placebo in preventing slow labor progress in a population of first-time mothers at risk of prolonged labor. Further research is warranted to answer whether increased and/or repeated doses of hyoscine butylbromide might have an effect on duration of labor.

## Trial registration

ClinicalTrials.gov (NCT03961165) EudraCT (2018-002338-19)

## Author summary

### Why was this study done?

➤ Approximately 10% of first-time mothers experience prolonged labor, which is associated with increased risk of operative delivery, postpartum hemorrhage, perineal tears, infections, and transfer to the neonatal intensive care unit (NICU).

➤ Intravenous synthetic oxytocin is widely used to treat slow labor progress, but the treatment is potentially harmful. Augmentation of contractions may lead to uterine hyperstimulation, which is associated with fetal distress and operative delivery. Hence, assessment of other treatments is needed.

➤ Previous research demonstrates that spasmolytics may shorten duration of labor, but studies of the effect of this treatment on preventing prolonged labor are lacking.

**What did the researchers do and find?**

➢ We performed a double-blind randomized placebo-controlled study including 249 nulliparous women showing first signs of slow labor progress.

➢ The participants were randomized to receive either a single intravenous dose of the spasmolytic drug hyoscine butylbromide (Buscopan) (20 mg) or a single intravenous dose of saline solution (placebo).

➢ We found no statistically significant difference in labor duration between the 2 treatment groups. There was a decrease in postpartum hemorrhage and a slight increase in maternal heart rate in the hyoscine butylbromide group.

➢ No maternal serious adverse events were observed nor did we observe any neonatal adverse events.

**What do these findings mean?**

➢ Hyoscine butylbromide was not found to impact duration of labor from treatment administration to delivery for first-time mothers with long labors.

➢ Hyoscine butylbromide can safely be used during labor.

➢ The effect of hyoscine butylbromide is short-lasting; a single intravenous dose of 20 mg might not have been sufficient to shorten labor in this selected group of nulliparous women.

➢ Randomized controlled trials (RCTs) assessing the effect of higher or repeated doses of hyoscine butylbromide for women experiencing long labors are needed.

## Introduction

Slow progress in labor is a common challenge in obstetric care, especially among first-time mothers (nulliparous women). Slow labor might lead to prolonged labor, most commonly defined as by the World Health Organization (WHO): an active labor lasting >12 h [1]. Prolonged labor causes as many as 2 out of 3 unplanned cesarean sections [2,3]. In addition to increased risk of operative delivery, prolonged labor is associated with chorioamnionitis [4], a negative birth experience [5,6], shoulder dystocia [7], low Apgar scores [7], and admission to the neonatal intensive care unit (NICU) [4,7]. There are also increased risks of severe postpartum hemorrhage [8], anal incontinence [9], urinary retention [10], hematomas [10], and ruptured sutures [10]. Hence, prolonged labor is a major clinical problem in obstetrics that needs to be identified and treated.

When progress in labor is too slow, augmentation with synthetic oxytocin is commonly used to increase contractions [11]. However, oxytocin has an unpredictable therapeutic index and has been described as the drug most commonly related to preventable adverse events during labor [11,12]. Treatment with oxytocin might be associated with increased risks of fetal asphyxia [13–15], operative delivery [15–17], a negative birth experience [15,16,18], anal sphincter injuries [19], postpartum urinary retention [20], postpartum hemorrhage [21], and

delayed initiation of breastfeeding [16,22–24]. Given the uncertainty regarding efficacy of oxytocin and the considerable side effects, there is a need to evaluate alternative adjuvant treatments to prevent prolonged labor.

Antispasmodics are drugs that relieve spasms of smooth muscle tissue, breaking the connection between the parasympathetic nerves and the smooth muscle by acting as antagonists of acetylcholine at muscarinic receptors and thereby inhibiting muscle spasms [25]. In general medicine, the drug is used for reducing spasms in the intestines and urinary tract, abdominal pain of unclear cause, irritable bowel disease, advanced cancer, colonoscopy, radiological examinations, and spasmodic conditions in the female genitalia [26]. The cervix is composed of connective tissue and smooth muscle which constitutes about 50% to 60% of the internal cervix and innervated by parasympathetic nerve fibers [27]. Musculotropic antispasmodics like hyoscine butylbromide (equivalent to butylscopamine bromide, Buscopan) directly relax smooth muscles [27]. Antispasmodics are most commonly used during labor in low- and middle-income countries [25]. Hyoscine butylbromide is an antispasmodic drug with a rapid onset (<20 min) [28]. A Cochrane review including 17 randomized controlled trials (RCTs) on the use of spasmolytics and duration of labor found a statistically significant reduction in the mean duration of the first stage of labor (i.e., the dilation phase) of 74 min [25]. A systematic review including 20 RCTs and a meta-analysis including 9 RCTs found a reduction in the mean duration of labor of 58 min and 55 min, respectively, when nulliparous women were treated with hyoscine butylbromide [29,30]. To our knowledge, except from 1 study from Saudi Arabia [31], there are no studies from high-income countries exploring the effect of hyoscine butylbromide on labor duration. Previous studies have been conducted in India, Pakistan, Iran, Iraq, Turkey, Nigeria, Mexico, Egypt, and Jamaica [29,30,32]. We have only found 1 Egyptian study on the effect of hyoscine butylbromide on slow labor [33].

In contrast to previous research on the effect of hyoscine butylbromide on labor duration in general, the double-blind randomized placebo-controlled trial BUSCLAB (BUSCopan in LABor) aimed to assess the effect of hyoscine butylbromide on the duration of the active phase of labor in nulliparous women showing early signs of slow labor.

## Methods

This study is reported as per the Consolidated Standards of Reporting Trials (CONSORT) guidelines (S1 CONSORT Checklist).

### Study design

We conducted a double-blind randomized placebo-controlled trial in a tertiary hospital in Norway with around 2,500 births annually. Further detailed information on the design may be found in the published protocol article [34]. The study protocol, including the sample size calculations, is provided in S1 Text.

The study was approved by the Regional Committee for Medical and Health Research Ethics South East Norway (2018/2380) and the Norwegian Medicines Agency (18/09179-14) and was registered in ClinicalTrials.gov (NCT03961165) and EudraCT (2018-002338-19) prior to inclusion of the first participant. The study was conducted in compliance with the Declaration of Helsinki and the International Conference on Harmonization Good Clinical Practice Guideline.

### Participants

In the period from May 2019 to July 2021, all nulliparous women scheduled to give birth at Oslo University Hospital Rikshospitalet were sent an information-and-invitation letter to

participate in the study approximately 6 weeks prior to their due date. A total of 1,900 women were invited.

## Inclusion criteria

Women aged 18 years or older with a singleton fetus in a cephalic presentation at term were potentially eligible for participation in the study. The participants had to be in active labor, i.e., having regular contractions and a cervical dilation of at least 3 cm. Start of labor had to be spontaneous, and progress of labor had to be slower than 1 cm per hour, as indicated by the WHO partograph (Fig 1).

The WHO partograph is a chart widely used for measuring labor progress and contains 2 diagonal lines: an alert line and an action line [35]. The alert line corresponds to an average dilation rate of 1 cm per hour. If the labor curve crosses to the right of this alert line, it means that the dilation is less than 1 cm per hour and may indicate early signs of slow labor progress. The action line starts 4 h to the right of the alert line. If the labor curve crosses the action line, intravenous infusion of oxytocin should be started to augment labor. In case of intact membranes, amniotomy should be performed and infusion of oxytocin started after 1 h if slow progress of labor persists [11].

We aimed to include all eligible women who crossed the alert line of the WHO partograph to assess whether a single intravenous dose of 1 ml hyoscine butylbromide (20 mg/ml) could reduce duration of labor in a group of women showing first signs of prolonged labor. All participating women provided written informed consent, signed by both the woman and the midwife in charge of her. As the participants were informed about the study in a letter sent to them during pregnancy, some women had signed the consent form before admission to hospital, confirming that they would participate provided labor would be slow and the inclusion criteria for participation were fulfilled. Other participants signed after admission to labor and delivery. All midwives in the labor ward were trained in the study procedures and could provide oral information when needed.

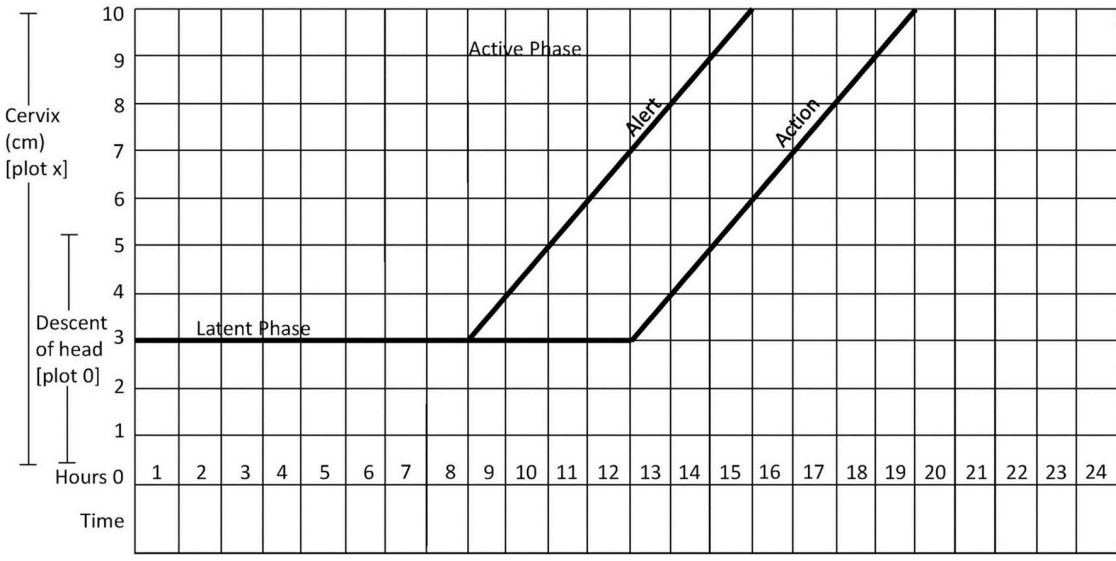

**Fig 1. WHO partograph in management of labor (26).** Reprinted with permission from Professor Tina Lavender. WHO, World Health Organization.

## Exclusion criteria

Women were excluded from participation in the study if they met any of the following exclusion criteria: treatment with synthetic oxytocin before start of active labor, preeclampsia, previous major uterine surgery, intestinal stenosis, ileus or megacolon, persisting maternal tachycardia (heart rate >130 beats/min), myasthenia gravis, untreated glaucoma, hypersensitivity to hyoscine butylbromide, maternal heart disease in need of telemetry surveillance during labor, persisting fetal tachycardia (fetal baseline heart rate >170 beats/min), or known fetal heart disease. Antihistamines, tricyclic and tetracyclic antidepressants, antipsychotic drugs, tiotropium, ipratropium, atropine, metoclopramide, and beta-adrenergic medications should not be administered with hyoscine butylbromide, and administration of any of these medications was regarded as an exclusion criterion.

## Randomization and blinding

The allocation sequence based on computer-generated random numbers was generated by an independent statistician who was not part of the trial and did not take part in the statistical analyses. Participants were assigned to either the study drug or placebo by block randomization with mixed block sizes of 2, 4, and 6. The treatment was concealed in sealed opaque envelopes with consecutive numbers.

Midwives at the labor and delivery ward enrolled eligible participants by the abovementioned inclusion and exclusion criteria. The participants were assigned to treatment based on the prespecified allocation written in the envelopes. The envelopes were kept in a locked room at the postnatal ward, situated on a different floor from the labor and delivery ward.

An authorized midwife at the postnatal ward opened the envelope and revealed the allocated treatment. This midwife had no other involvement in the study and was the only person unblinded to which treatment the participant received. The envelope was labeled with the participant's name and date of birth and resealed with a special blue tape that would leave colored traces if reopening had been attempted. After opening the envelope, the midwife prepared a syringe with either 1 ml hyoscine butylbromide (20 mg/ml) or 1 ml sodium chloride (9 mg/ml, 0.9%). In case of a serious adverse event, unblinding was done by opening the woman's envelope. All packaging identifying the allocated treatment was discarded. The 2 treatments had identical appearance. The midwife gave the syringe to a third person, blinded to its content. This third person brought the syringe to the midwife in charge of the study participant at the labor and delivery ward.

The investigational medicinal product (IMP), either hyoscine butylbromide or sodium chloride (placebo), was administered to the woman by authorized site personnel only. Authorization was given by the principal investigator (PI) after oral and written information and individual training in the study procedures. The IMP had to be given within 45 min after the vaginal examination showing that the woman was eligible (i.e., had crossed the alert line of the WHO partograph). Fetal cardiotocography (CTG) lasting at least 10 min was performed before IMP was given and 30 min post administration. Furthermore, maternal heart rate was measured before and 30 min after IMP administration. All participants were given standard care as per ward procedure.

## Procedures following treatment administration

Occurrence, degree, and duration of adverse effects were registered 30 min after the IMP was given and daily at the postnatal ward, see S1 Fig and the protocol article for this study [34]. Measurements of the pH levels of arterial and venous umbilical cord blood were obtained

immediately after delivery. The Apgar score of the newborn was also assessed 5 and 10 min after birth.

Infant pulse oximetry with $O_2$ saturation was performed routinely when the neonate was approximately 4 h old. Physical examination was performed on the first or second day postpartum by an experienced pediatrician. All infants were observed as per ward procedure by trained personnel at the postnatal ward throughout their stay. Physical examination of the mother included measurements of the heart rate and blood pressure at least twice a day during the hospital stay.

All women and infants were followed for adverse events from randomization to discharge from the hospital up to 3 days after delivery, in accordance with clinical practice.

### Baseline characteristics

Baseline characteristics included maternal age (in years), height (in cm), weight (in kg), body mass index (BMI) (in kg/m$^2$), marital status (married, cohabiting, single, or unknown), higher education (yes or no), chronic diseases (yes or no), and use of medication during pregnancy (yes or no) in addition to maternal and fetal heart rates at baseline (in beats/min) and cervical dilation at baseline (in cm). Maternal age was categorized as <25 years, 25 to 29 years, 30 to 34 years, and ≥35 years, and maternal BMI was categorized according to the WHO's definitions of underweight (BMI: <18.5 kg/m$^2$), normal weight (BMI: 18.5 to 24.9 kg/m$^2$), overweight (BMI: 25.0 to 29.9 kg/m$^2$), and obesity (BMI: ≥30.0 kg/m$^2$).

### Outcomes

We evaluated a total of 23 outcomes in the present study: 3 time-to-event outcomes, 9 continuous outcomes, and 11 dichotomous outcomes.

The primary outcome of the study was duration of labor from IMP administration to vaginal delivery (in minutes), which is a time-to-event variable, with vaginal delivery as the event of interest and emergency cesarean section as a competing event.

Secondary time-to-event outcomes included duration of labor from onset of active labor to vaginal delivery (in minutes), with emergency cesarean section as a competing event, and duration of labor from IMP administration to a fully dilated cervix of 10 cm, with emergency cesarean section prior to a cervical dilation of 10 cm as a competing event.

Secondary continuous outcomes included cervical dilation rates (in cm per hour) from IMP administration to a fully dilated cervix and from IMP administration to the next vaginal examination, changes in maternal and fetal heart rates 30 min after IMP administration (in beats/min), duration of oxytocin infusion (in minutes), amount of infused oxytocin (in international units [IU]), estimated blood loss (in ml), and pH levels of the umbilical artery and vein.

Secondary dichotomous outcomes included operative vaginal delivery, emergency cesarean section, postpartum hemorrhage ≥500 ml, ≥1,000 ml, and ≥1,500 ml, respectively, pH level of the umbilical artery <7.0, pH level of the umbilical vein <7.1, 5- and 10-min Apgar scores <7, and admission of the newborn to the NICU within the first couple of hours after birth.

### Sample size calculations

The sample size calculations were based on an RCT by Dencker and colleagues from 2009, which found that among first-time mothers with spontaneous start of labor, the mean duration of labor from randomization to delivery was 5.2 h, with a standard deviation (SD) of 2.8 h [36]. A difference in means of 60 min in labor duration was considered clinically relevant, as all the other studies on spasmolytics found a difference of more than 55 min. With a statistical

power of 80% and an SD of 2.8 as given in Dencker and colleagues, our sample size calculations showed that we would require 246 women in total to discover a difference in duration of labor of 60 min between the 2 treatment groups. More details on the sample size calculations may be found in the study protocol (S1 Text) and the statistical analysis plan (S2 Text, Section 2.3).

### Analysis sets

We considered 4 different analysis sets. The restricted intention-to-treat (ITT) set comprised all eligible, randomized trial participants with a signed informed consent, regardless of protocol adherence. The full-analysis (FA) set comprised all trial participants in the restricted ITT set who received the IMP, thereby excluding trial participants not given the IMP due to reasons unrelated to the allocated treatment. The FA set was thus regarded as a modified ITT set for which the ITT principle was still preserved. The per-protocol (PP) set comprised all trial participants in the FA set with no major protocol deviations (S2 Text, Section 3.2.2). Finally, the safety set comprised all randomized trial participants who ever received the IMP, irrespective of eligibility.

From May 2019 to December 2021, 1,900 women were assessed for eligibility. A total of 261 women were initially randomized to treatment. Ten of the randomized women were subsequently excluded due to fullfilling one of the exclusion criteria, and 1 randomized woman later withdrew her consent, leaving 250 women in the restricted ITT set (126 in the hyoscine butylbromide group and 124 in the placebo group). Furthermore, 1 woman in the hyoscine butylbromide group did not receive the IMP, and the FA set therefore comprised 249 women (125 in the hyoscine butylbromide group and 124 in the placebo group). The PP set included 248 women (124 in the hyoscine butylbromide group and 124 in the placebo group), excluding 1 woman in the hyoscine butylbromide group whose midwife was unblinded to the IMP administered. The safety set included 255 women: the 249 women in the FA set in addition to 6 non-eligible, erroneously randomized women who received 1 dose of the IMP. The CONSORT flow diagram of the enrollment of trial participants is shown in Fig 2.

Descriptive statistics of baseline characteristics of the women in the FA set, overall and by treatment group, are given in Table 1. There were no signs of imbalance between the 2 treatment groups with respect to any of the baseline characteristics other than slight differences in the continuous variables for maternal weight (Mann–Whitney $U$ test: $p = 0.028$) and BMI (Mann–Whitney $U$ test: $p = 0.009$). Still, we found no statistically significant association between categorical maternal BMI and treatment group.

### Statistical analysis

All statistical analyses were prespecified in the statistical analysis plan (S2 Text), which was dated and signed by the PI and the trial statistician prior to the final database lock and unblinding of the treatment allocation of all trial participants.

The analyses of the outcomes were primarily based on the FA set. Sensitivity analyses similar to the main analyses based on the PP set were conducted for comparison purposes. Furthermore, analyses of the cervical dilation rate from IMP administration to a fully dilated cervix of 10 cm were restricted to trial participants with a cervical dilation of 10 cm prior to delivery, and analyses of operative vaginal delivery were restricted to trial participants with vaginal delivery. Since we observed no imbalance in relevant baseline characteristics between the 2 treatment groups in the FA set (Table 1), all analyses were crude except for the analyses of the changes in maternal and fetal heart rates 30 min after IMP administration, which were adjusted for maternal and fetal heart rates at baseline, respectively (S2 Text, Section 5.2).

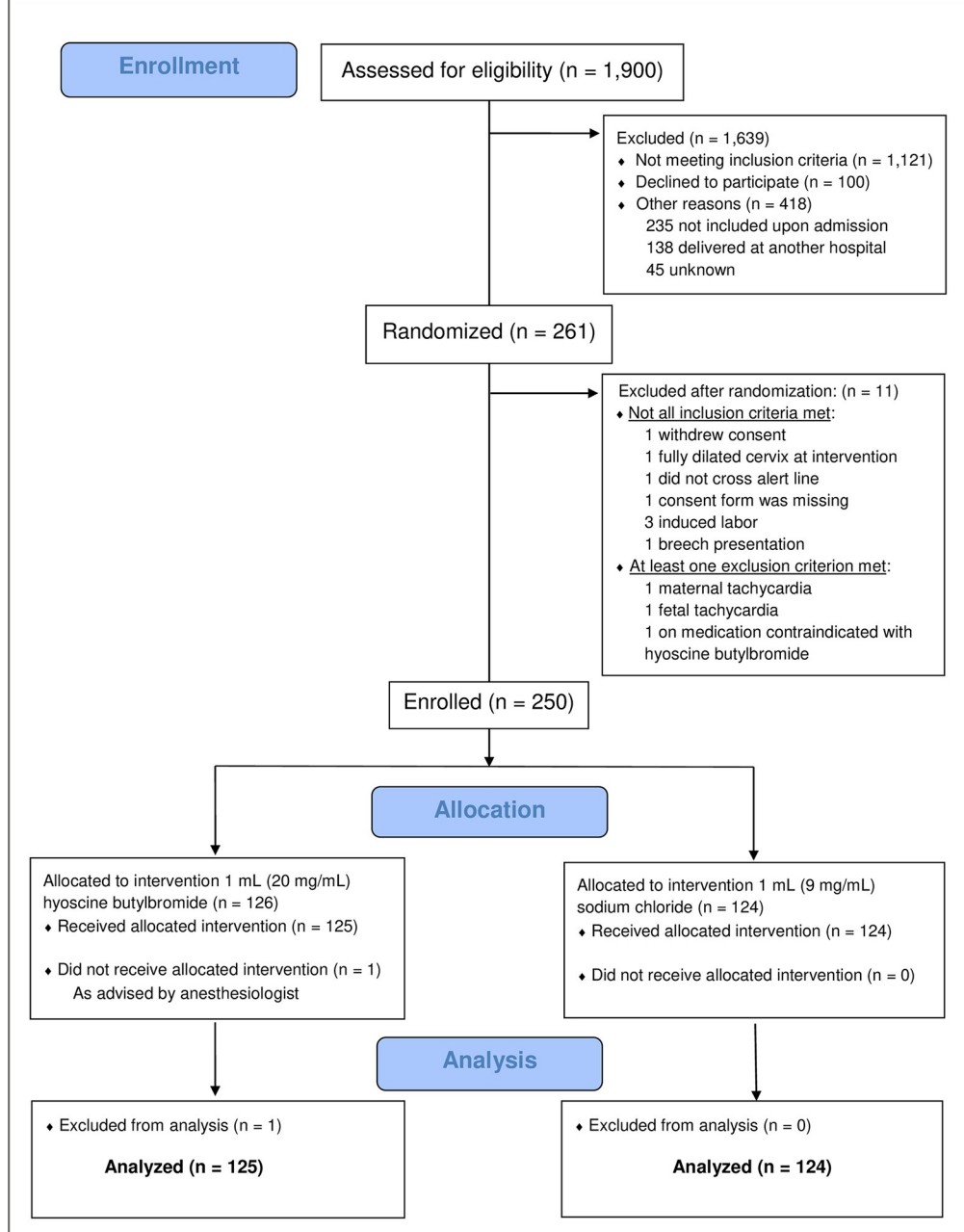

**Fig 2. CONSORT flow diagram of the BUSCLAB (BUSCopan in LABor) study.**

Furthermore, we did not correct for multiple testing in any of the analyses, as there was only a single primary outcome in the present study. All efficacy analyses of the secondary outcomes were therefore regarded as supportive.

We used Weibull regression to analyze duration of labor from IMP administration to vaginal delivery (primary outcome). The choice of such a parametric survival model would potentially increase the statistical power of detecting a real difference between the treatment groups, given that the event times follow a Weibull distribution. We estimated the cause-specific hazard ratio (HR) of vaginal delivery between the hyoscine butylbromide group and the placebo

**Table 1. Descriptive statistics of baseline characteristics, by treatment group, based on the FA set.** Continuous variables are given by mean (SD) (normal data) or median (IQR) (non-normal data). Categorical variables are given by numbers (percentages).

| Baseline characteristic | Total (*n* = 249) | | Hyoscine butylbromide (*n* = 125) | | Placebo (*n* = 124) | |
|---|---|---|---|---|---|---|
| *Maternal age (in years)* | | | | | | |
| Mean (SD) | 32.0 | (4.0) | 31.7 | (4.4) | 32.3 | (3.6) |
| *Maternal age (in years)* | | | | | | |
| <25 | 8 | (3.2%) | 6 | (4.8%) | 2 | (1.6%) |
| 25–29 | 55 | (22.1%) | 31 | (24.8%) | 24 | (19.4%) |
| 30–34 | 124 | (49.8%) | 59 | (47.2%) | 65 | (52.4%) |
| ≥35 | 62 | (24.9%) | 29 | (23.2%) | 33 | (26.6%) |
| *Maternal height (in cm)* | | | | | | |
| Mean (SD) | 167.1 | (6.7) | 167.0 | (6.6) | 167.2 | (6.8) |
| *Maternal weight (in kg)* | | | | | | |
| Median (IQR) | 62.0 | (12.0) | 61.0 | (11.0) | 63.0 | (14.0) |
| *Maternal BMI (in kg/m$^2$)* | | | | | | |
| <18.5 | 9 | (3.6%) | 7 | (5.6%) | 2 | (1.6%) |
| 18.5–24.9 | 188 | (75.5%) | 97 | (77.6%) | 91 | (73.4%) |
| 25.0–29.9 | 39 | (15.7%) | 16 | (12.8%) | 23 | (18.5%) |
| ≥30.0 | 13 | (5.2%) | 5 | (4.0%) | 8 | (6.5%) |
| *Maternal marital status* | | | | | | |
| Married | 87 | (34.9%) | 47 | (37.6%) | 40 | (32.3%) |
| Cohabiting | 150 | (60.2%) | 70 | (56.0%) | 80 | (64.5%) |
| Single | 10 | (4.0%) | 7 | (5.6%) | 3 | (2.4%) |
| Unknown | 2 | (0.8%) | 1 | (0.8%) | 1 | (0.8%) |
| *Maternal higher education* | | | | | | |
| No | 13 | (5.2%) | 4 | (3.2%) | 9 | (7.3%) |
| Yes | 234 | (94.0%) | 119 | (95.2%) | 115 | (92.7%) |
| Missing | 2 | (0.8%) | 2 | (1.6%) | 0 | (0.0%) |
| *Maternal chronic diseases* | | | | | | |
| No | 157 | (63.1%) | 82 | (65.6%) | 75 | (60.5%) |
| Yes | 91 | (36.5%) | 43 | (34.4%) | 48 | (38.7%) |
| Missing | 1 | (0.4%) | 0 | (0.0%) | 1 | (0.8%) |
| *Maternal medication during pregnancy* | | | | | | |
| No | 150 | (60.2%) | 82 | (65.6%) | 68 | (54.8%) |
| Yes | 99 | (39.8%) | 43 | (34.4%) | 56 | (45.2%) |
| *Maternal pain score at baseline* | | | | | | |
| Median (IQR) | 3.0 | (5.0) | 3.0 | (5.0) | 3.0 | (5.0) |
| Missing | 1 | (0.4%) | 0 | (0.0%) | 1 | (0.8%) |
| *Maternal heart rate at baseline (in beats/min)* | | | | | | |
| Mean (SD) | 86.5 | (14.9) | 87.5 | (15.7) | 85.4 | (13.9) |
| *Fetal heart rate at baseline (in beats/min)* | | | | | | |
| Mean (SD) | 139.8 | (10.5) | 139.6 | (10.2) | 139.9 | (10.9) |
| *Cervical dilation at baseline (in cm)* | | | | | | |
| Median (IQR) | 5.0 | (2.0) | 5.0 | (2.0) | 5.0 | (2.0) |

BMI, body mass index; FA, full analysis; IQR, interquartile range; SD, standard deviation.

group, with associated 95% confidence interval (CI), treating emergency cesarean section as a censoring event. Correspondingly, we estimated the cause-specific HR of emergency cesarean section, treating vaginal delivery as a censoring event. We also calculated the respective

cumulative incidences of vaginal delivery and emergency cesarean section by treatment group. Cox regression was applied in a sensitivity analysis to check the robustness of the results of the main analysis. To explore a potential treatment effect modification by labor progression before IMP administration, we conducted subgroup analyses similar to the main analysis by considering the interaction between IMP and cervical dilation before IMP administration (<5 cm or ≥5 cm).

Secondary time-to-event outcomes were analyzed in the same way as the primary outcome.

We analyzed the secondary continuous outcomes by using median regression to estimate the difference in medians between the 2 treatment groups, as the conditions for linear regression were not met.

Secondary dichotomous outcomes were analyzed by using binary logistic regression, estimating odds ratios (ORs) between the 2 treatment groups. Pearson's chi-square test or Fisher's exact test, as appropriate, was also performed in a sensitivity analysis of each outcome.

There were no missing data on any of the outcomes except for changes in maternal and fetal heart rates due to missing values of the heart rates 30 min after IMP administration (both 0.4%), pH level of the umbilical artery (32.9%), pH level of the umbilical vein (13.3%), and 10-min Apgar score (2.4%).

We used mean (single) imputation, performed separately within each treatment group, to handle missing data on the maternal and fetal heart rates 30 min after IMP administration before the respective changes were calculated. Missing data on the pH levels of the umbilical artery and vein were handled by using multiple imputation, including allocated treatment, categorical maternal age, and categorical maternal BMI in the imputation model. The multiple imputation was performed separately for the 2 pH levels in the main analyses, whereas a mutual imputation process for both pH levels was used in sensitivity analyses. To handle missing data on the 10-min Apgar score, we used best-case (single) imputation, with 10-min Apgar score ≥7 defined as the best-case scenario. Worst-case (single) imputation of this outcome was used in a sensitivity analysis. For all outcomes with missing values, complete-case analyses were also conducted as sensitivity analyses.

Further details regarding the statistical analysis are provided in S2 Text.

## Monitors

The study was supervised by an external clinical study monitor from the Clinical Trials Unit at Oslo University Hospital. The monitor performed reviews every 6 months throughout the study period. The clinical data managers and the study monitor remotely and proactively monitored the electronic case report forms (eCRFs) to improve the data quality.

## Results

Out of 1,900 first-time mothers invited during pregnancy, 1,121 did not meet the inclusion criteria, 138 delivered at another hospital, and 45 did not participate for reasons unknown. Of the 596 eligible women, 100 did not wish to be included and 235 were not included upon admission. The remaining 261 were included in the study, 11 of whom were later excluded because they did not fulfill the inclusion criteria (Fig 2).

Numbers and percentages of women giving birth within 2, 4, 6, 8, 10, and 12 h after IMP administration, by treatment group and mode of delivery (vaginal delivery or emergency cesarean section), are displayed in Fig 3. The proportion of women giving birth within 6 h after IMP administration was higher in the hyoscine butylbromide group compared to the placebo group (42.1% versus 30.7%).

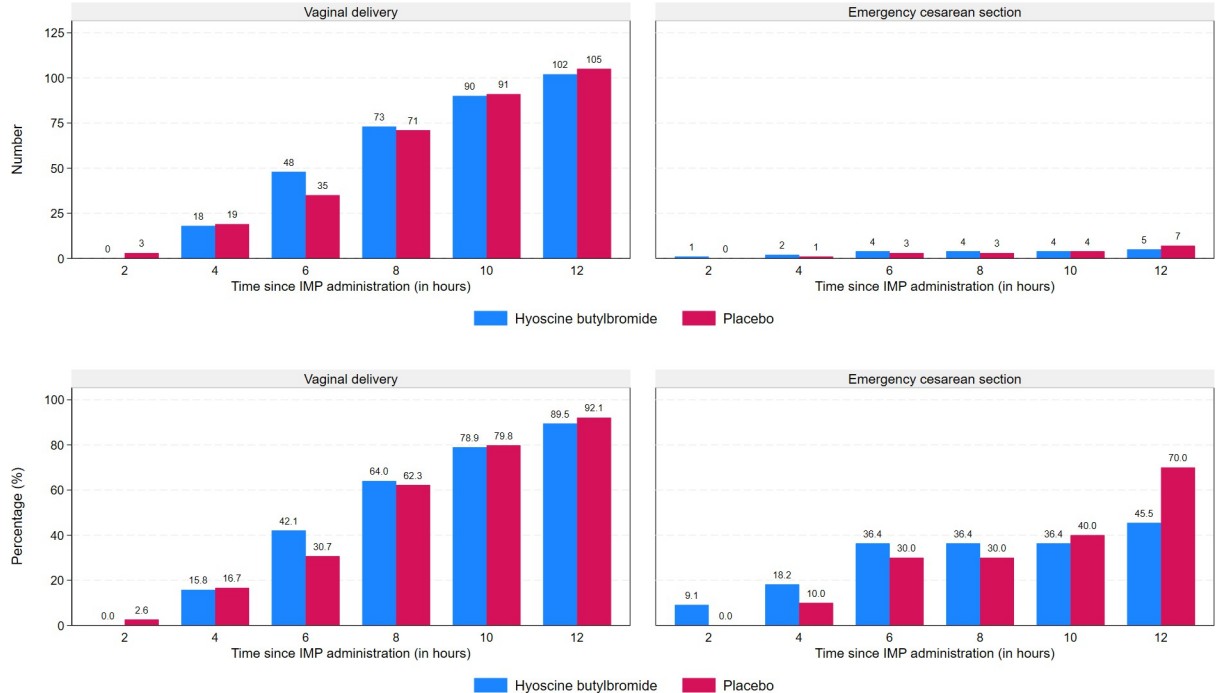

**Fig 3. Absolute and relative frequencies of deliveries as functions of time since administration of the IMP for 1 ml (20 mg) hyoscine butylbromide and 1 ml (9 mg) sodium chloride, respectively.** Emergency cesarean section was defined as cesarean section of which an indication developed during labor. IMP, investigational medicinal product.

Descriptive statistics and results of the main analyses of the outcomes under study are presented in Table 2. Results of all analyses prespecified in the statistical analysis plan, including sensitivity analyses and subgroup analyses, may be found in S1 Table. Furthermore, S2 Table presents the results of post hoc adjusted sensitivity analyses where the main analyses were adjusted for categorical maternal age at baseline (all outcomes), categorical maternal BMI at baseline (all outcomes), and cervical dilation at baseline (3 outcomes related to duration of labor and 2 outcomes related to cervical dilation rate only).

Overall, the observed median (mean) duration of labor from IMP administration to delivery, irrespective of mode of delivery, was 412.0 (459.7) min in the hyoscine butylbromide group and 436.0 (464.3) min in the placebo group. The corresponding numbers restricted to vaginal deliveries were 401.0 (440.8) min and 432.5 (453.6) min, respectively (Table 2). We found no evidence of an effect of hyoscine butylbromide on duration of labor from IMP administration to vaginal delivery, with a cause-specific HR of vaginal delivery of 1.00 (95% CI [0.77, 1.29]; $p$ = 0.993) (Table 2). Correspondingly, the cause-specific HR of emergency cesarean section was 1.11 (95% CI [0.47, 2.62; $p$ = 0.811) (S1 Table). The assumptions of proportional hazards and Weibull-distributed baseline hazard rate were evaluated and found valid. Results of the sensitivity analysis in which Cox regression was applied were similar to those of the main analysis (S1 Table). Curves of the respective cause-specific hazard rates of vaginal delivery and emergency cesarean section as functions of time since IMP administration are displayed in Fig 4. Fig 5 displays curves of the respective cumulative incidences of vaginal delivery and emergency cesarean section as functions of time since IMP administration.

Results of a post hoc competing-risks regression analysis of time from IMP administration to vaginal delivery were similar to the results of the main analysis, with a sub-hazard ratio (SHR) of 1.00 (95% CI [0.77, 1.30]; $p$ = 0.990).

**Table 2. Descriptive statistics of outcomes, by treatment group, and results of the main statistical analyses, based on the FA set.**

| Outcome | Total (n = 249) | | Hyoscine butylbromide (n = 125) | | Placebo (n = 124) | | Effect measure | Point estimate | 95% CI | p value |
|---|---|---|---|---|---|---|---|---|---|---|
| *Duration of labor from IMP administration to vaginal delivery (in min)* | | | | | | | HR[5] | 1.00 | [0.77, 1.29] | 0.993 |
| Median (IQR) | 429.0 | (279.5) | 401.0 | (304.0) | 432.5 | (262.0) | | | | |
| Censoring[1] | 21 | (8.4%) | 11 | (8.8%) | 10 | (8.1%) | | | | |
| *Duration of labor from onset of active labor to vaginal delivery (in min)* | | | | | | | HR[5] | 1.02 | [0.79, 1.32] | 0.893 |
| Median (IQR) | 720.5 | (293.5) | 727.0 | (318.0) | 712.5 | (268.0) | | | | |
| Censoring[1] | 21 | (8.4%) | 11 | (8.8%) | 10 | (8.1%) | | | | |
| *Duration of labor from IMP administration to cervical dilation of 10 cm (in min)* | | | | | | | HR[6] | 0.93 | [0.72, 1.20] | 0.555 |
| Median (IQR) | 308.0 | (255.0) | 291.0 | (258.0) | 314.0 | (264.0) | | | | |
| Censoring[2] | 15 | (6.0%) | 10 | (8.0%) | 5 | (4.0%) | | | | |
| *Cervical dilation rate from IMP administration to cervical dilation of 10 cm (in cm/hour)[3]* | | | | | | | DM | 0.04 | [−0.16, 0.25] | 0.683 |
| Median (IQR) | 0.9 | (0.9) | 0.9 | (0.9) | 0.9 | (0.9) | | | | |
| *Cervical dilation rate from IMP administration to first vaginal examination after IMP administration (in cm/hour)* | | | | | | | DM | 0.10 | [−0.31, 0.51] | 0.626 |
| Median (IQR) | 0.7 | (1.3) | 0.7 | (1.4) | 0.6 | (1.3) | | | | |
| *Change in maternal heart rate (in beats/min)* | | | | | | | DM[7] | 3.03 | [0.35, 5.71] | 0.027 |
| Median (IQR) | 4.0 | (14.5) | 5.0 | (17.0) | 2.0 | (12.0) | | | | |
| Missing | 1 | (0.4%) | 0 | (0.0%) | 1 | (0.8%) | | | | |
| *Change in fetal heart rate (in beats/min)* | | | | | | | DM[8] | 0.00 | [−1.52, 1.52] | 1.000 |
| Median (IQR) | 0.0 | (9.5) | 0.0 | (10.0) | 0.0 | (8.0) | | | | |
| Missing | 1 | (0.4%) | 0 | (0.0%) | 1 | (0.8%) | | | | |
| *Duration of oxytocin infusion (in min)* | | | | | | | DM | 49.00 | [−50.74, 148.74] | 0.334 |
| Median (IQR) | 217.0 | (361.0) | 247.0 | (324.0) | 196.0 | (400.5) | | | | |
| *Amount of infused oxytocin (in IU)* | | | | | | | DM | 0.24 | [−0.52, 1.00] | 0.537 |
| Median (IQR) | 1.4 | (3.4) | 1.6 | (2.7) | 1.3 | (3.9) | | | | |
| *Operative delivery* | | | | | | | OR | 0.89 | [0.54, 1.48] | 0.661 |
| No | 142 | (57.0%) | 73 | (58.4%) | 69 | (55.6%) | | | | |
| Yes | 107 | (43.0%) | 52 | (41.6%) | 55 | (44.4%) | | | | |
| *Operative vaginal delivery[4]* | | | | | | | OR | 0.86 | [0.50, 1.47] | 0.585 |
| No | 142 | (62.3%) | 73 | (64.0%) | 69 | (60.5%) | | | | |
| Yes | 86 | (37.7%) | 41 | (36.0%) | 45 | (39.5%) | | | | |
| *Emergency cesarean section* | | | | | | | OR | 1.10 | [0.45, 2.69] | 0.835 |
| No | 228 | (91.6%) | 114 | (91.2%) | 114 | (91.9%) | | | | |
| Yes | 21 | (8.4%) | 11 | (8.8%) | 10 | (8.1%) | | | | |
| *Estimated blood loss (in ml)* | | | | | | | DM | −50.00 | [−96.45, −3.55] | 0.035 |
| Median (IQR) | 400.0 | (200.0) | 400.0 | (200.0) | 435.0 | (325.0) | | | | |
| *Postpartum hemorrhage ≥500 ml* | | | | | | | OR | 0.66 | [0.39, 1.12] | 0.126 |
| No | 168 | (67.5%) | 90 | (72.0%) | 78 | (62.9%) | | | | |
| Yes | 81 | (32.5%) | 35 | (28.0%) | 46 | (37.1%) | | | | |
| *Postpartum hemorrhage ≥1,000 ml* | | | | | | | OR | 0.63 | [0.27, 1.47] | 0.285 |
| No | 224 | (90.0%) | 115 | (92.0%) | 109 | (87.9%) | | | | |
| Yes | 25 | (10.0%) | 10 | (8.0%) | 15 | (12.1%) | | | | |
| *Postpartum hemorrhage ≥1,500 ml* | | | | | | | OR | 0.13 | [0.02, 1.11] | 0.063 |
| No | 241 | (96.8%) | 124 | (99.2%) | 117 | (94.4%) | | | | |
| Yes | 8 | (3.2%) | 1 | (0.8%) | 7 | (5.6%) | | | | |
| *pH level of the umbilical artery* | | | | | | | DM | 0.01 | [−0.02, 0.04] | 0.514 |
| Median (IQR) | 7.2 | (0.1) | 7.2 | (0.1) | 7.2 | (0.1) | | | | |

*(Continued)*

**Table 2.** (Continued)

| Outcome | Total (*n* = 249) | | Hyoscine butylbromide (*n* = 125) | | Placebo (*n* = 124) | | Effect measure | Point estimate | 95% CI | *p* value |
|---|---|---|---|---|---|---|---|---|---|---|
| Missing | 82 | (32.9%) | 39 | (31.2%) | 43 | (34.7%) | | | | |
| *pH level of the umbilical artery <7.00* | | | | | | | OR | NA | NA | NA |
| No | 166 | (66.7%) | 86 | (68.8%) | 80 | (64.5%) | | | | |
| Yes | 1 | (0.4%) | 0 | (0.0%) | 1 | (0.8%) | | | | |
| Missing | 82 | (32.9%) | 39 | (31.2%) | 43 | (34.7%) | | | | |
| *pH level of the umbilical vein* | | | | | | | DM | −0.00 | [−0.02, 0.02] | 0.851 |
| Median (IQR) | 7.3 | (0.1) | 7.3 | (0.1) | 7.3 | (0.1) | | | | |
| Missing | 33 | (13.3%) | 12 | (9.6%) | 21 | (16.9%) | | | | |
| *pH level of the umbilical vein <7.10* | | | | | | | OR | NA | NA | NA |
| No | 216 | (86.7%) | 113 | (90.4%) | 103 | (83.1%) | | | | |
| Missing | 33 | (13.3%) | 12 | (9.6%) | 21 | (16.9%) | | | | |
| *5-min Apgar score <7* | | | | | | | OR | NA | NA | NA |
| No | 248 | (99.6%) | 124 | (99.2%) | 124 | (100.0%) | | | | |
| Yes | 1 | (0.4%) | 1 | (0.8%) | 0 | (0.0%) | | | | |
| *10-min Apgar score <7* | | | | | | | OR | NA | NA | NA |
| No | 243 | (97.6%) | 123 | (98.4%) | 120 | (96.8%) | | | | |
| Missing | 6 | (2.4%) | 2 | (1.6%) | 4 | (3.2%) | | | | |
| *Admission to the NICU* | | | | | | | OR | 1.33 | [0.29, 6.08] | 0.710 |
| No | 242 | (97.2%) | 121 | (96.8%) | 121 | (97.6%) | | | | |
| Yes | 7 | (2.8%) | 4 | (3.2%) | 3 | (2.4%) | | | | |

[1]Due to emergency cesarean section.

[2]Due to emergency cesarean section prior to cervical dilation of 10 cm.

[3]Restricted to 234 trial participants with cervical dilation of 10 cm prior to delivery.

[4]Restricted to 228 trial participants with vaginal delivery.

[5]Based on Weibull regression, with emergency cesarean section treated as a censoring event.

[6]Based on Weibull regression, with emergency cesarean section prior to cervical dilation of 10 cm treated as a censoring event.

[7]Adjusted for maternal heart rate at baseline.

[8]Adjusted for fetal heart rate at baseline.

CI, confidence interval; DM, difference in medians; FA, full analysis; HR, hazard ratio; IMP, investigational medicinal product; IQR, interquartile range; IU, international units; NA, not applicable; NICU, neonatal intensive care unit; OR, odds ratio; SD, standard deviation.

Similar to the analysis of the primary outcome, we did not find any statistically significant associations between treatment group and duration of labor from onset of active labor to vaginal delivery and duration of labor from IMP administration to a fully dilated cervix of 10 cm, respectively (Table 2 and S1 Table). The respective cumulative incidences of vaginal delivery and emergency cesarean section are plotted against time since onset of active labor in Fig 6. Fig 7 displays curves of the respective cumulative incidences of cervical dilation of 10 cm and emergency cesarean section prior to cervical dilation of 10 cm as functions of time since IMP administration.

SHRs obtained from post hoc competing-risks regression analyses of the secondary time-to-event outcomes were similar to the cause-specific HRs from the main analyses.

Change in maternal heart rate from IMP administration to 30 min after IMP administration was higher in the hyoscine butylbromide group than in the placebo group, with a difference in medians of 3.03 beats/min (95% CI [0.35, 5.71] beats/min; *p* = 0.027) (Table 2). Furthermore, estimated blood loss was slightly lower in the hyoscine butylbromide group

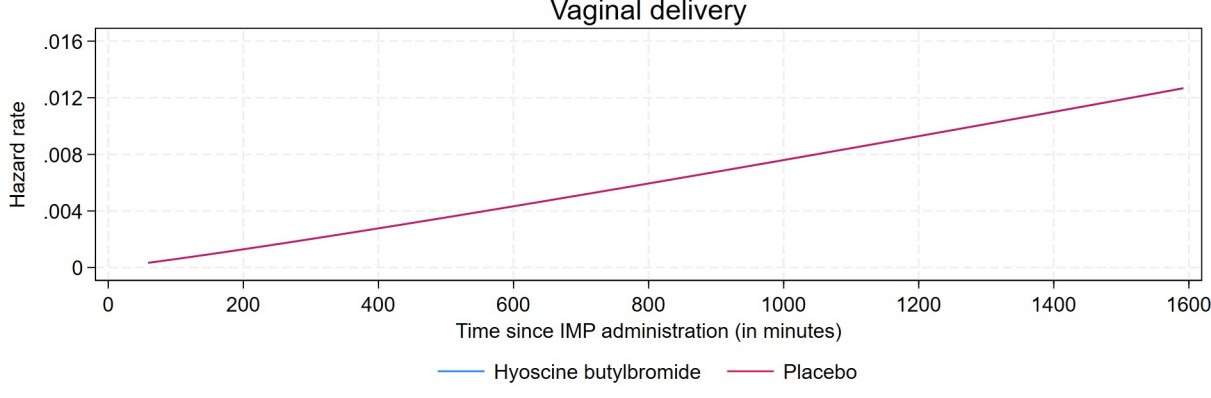

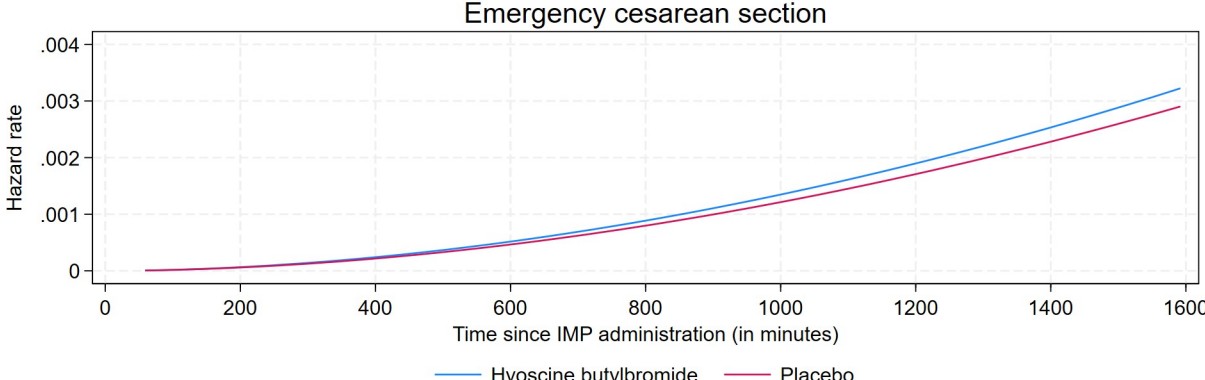

**Fig 4. Cause-specific hazard rates of vaginal delivery and emergency cesarean section as functions of time since administration of the IMP for 1 ml (20 mg) hyoscine butylbromide and 1 ml (9 mg) sodium chloride (placebo), respectively.** Emergency cesarean section was defined as cesarean section of which an indication developed during labor. The curves of the case-specific hazard rates of vaginal delivery of hyoscine butylbromide and placebo are completely overlapping. IMP, investigational medicinal product.

compared to the placebo group: difference in medians of −50.00 ml (95% CI [−96.45, −3.55] ml; $p = 0.035$) (Table 2). However, this association was no longer statistically significant in the post hoc adjusted sensitivity analysis including adjustment for categorical maternal age and categorical maternal BMI at baseline (S2 Table). We found no statistically significant associations between treatment group and any of the remaining secondary continuous outcomes (Table 2).

None of the secondary dichotomous outcomes were significantly associated with treatment group except for postpartum hemorrhage ≥1,500 ml (1 case in the hyoscine butylbromide group and 7 cases in the placebo group) in one of the sensitivity analyses (Fisher's exact test: $p = 0.036$ [S1 Table]). However, we observed no statistically significant association in the main analysis of this outcome, with an OR of 0.13 (95% CI [0.02, 1.11]; $p = 0.063$). Four of the outcomes (pH level of the umbilical artery <7.0, pH level of the umbilical vein <7.1, and 5- and 10-min Apgar scores <7) were not analyzed due to small numbers, i.e., cells with few or no observations (Table 2).

We observed 169 women (66.3%) in the safety set with any adverse event across all system organ classes (234 adverse events in total): 97 women (75.2%) in the hyoscine butylbromide group (146 adverse events in total) and 72 women (57.1%) in the placebo group (88 adverse events in total) (Pearson's chi-square test: $p = 0.002$). The numbers and percentages of women with adverse events are presented for both treatment groups by system organ class in Table 3.

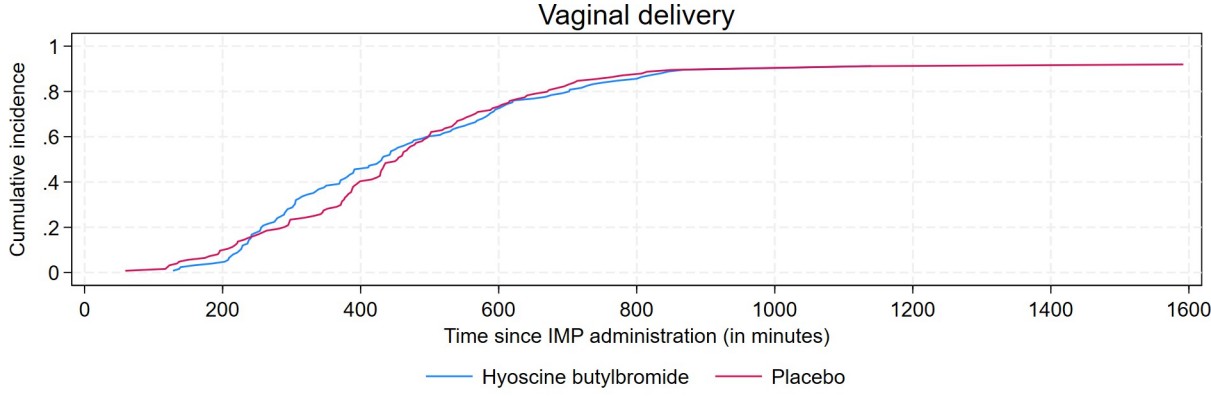

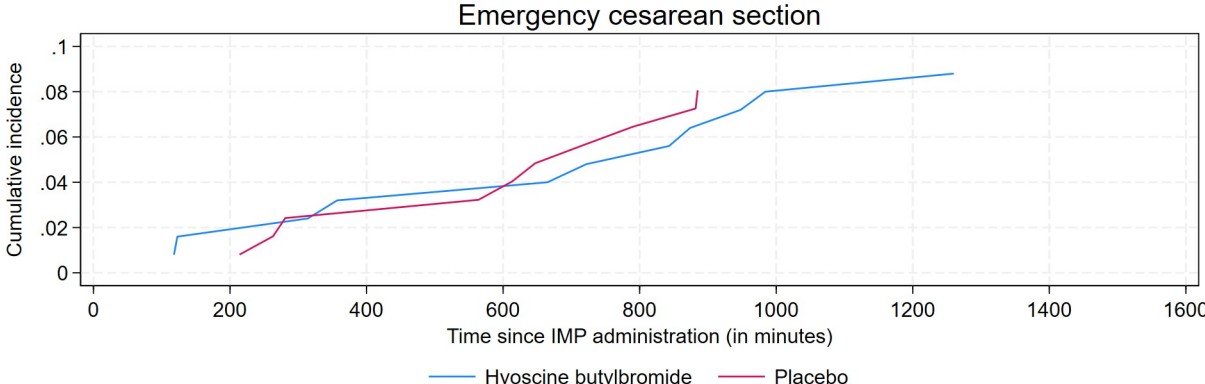

**Fig 5. Cumulative incidences of vaginal delivery and emergency cesarean section as functions of time since administration of the IMP for 1 ml (20 mg) hyoscine butylbromide and 1 ml (9 mg) sodium chloride, respectively.** Emergency cesarean section was defined as cesarean section of which an indication developed during labor. IMP, investigational medicinal product.

In the hyoscine butylbromide group, there was a higher number of reported tachycardia cases compared to the placebo group: 51 women (39.5%) versus 3 women (2.4%) (Pearson's chi-square test: $p < 0.001$). Similarly, a significant difference was observed in the occurrence of visual disturbances, with 8 women (6.2%) in the hyoscine butylbromide group and 1 woman (0.8%) in the placebo group (Fisher's exact test: $p = 0.036$). No fetal adverse events occurred during the trial nor were any maternal serious adverse events observed.

A more thorough analysis of the CTG as well as an evaluation of the birth experience will be provided in future publications.

## Discussion

By contrast with previous reports, we show no effect of hyoscine butylbromide on duration of labor from IMP administration to vaginal delivery. Furthermore, we found no associations between treatment group and duration of labor from IMP administration to a fully dilated cervix of 10 cm or from onset of active labor to vaginal delivery, nor did we find any differences between the treatment groups regarding cervical dilation rate. Administration of hyoscine butylbromide was associated with increased maternal heart rate 30 min after IMP administration and decreased postpartum hemorrhage; although, the differences between the 2 treatment groups were small and probably not clinically significant.

BUSCLAB strongly suggests that the efficacy of 20 mg hyoscine butylbromide in shortening labor in nulliparous women with signs of prolonged labor is either nonexistent or weak. To

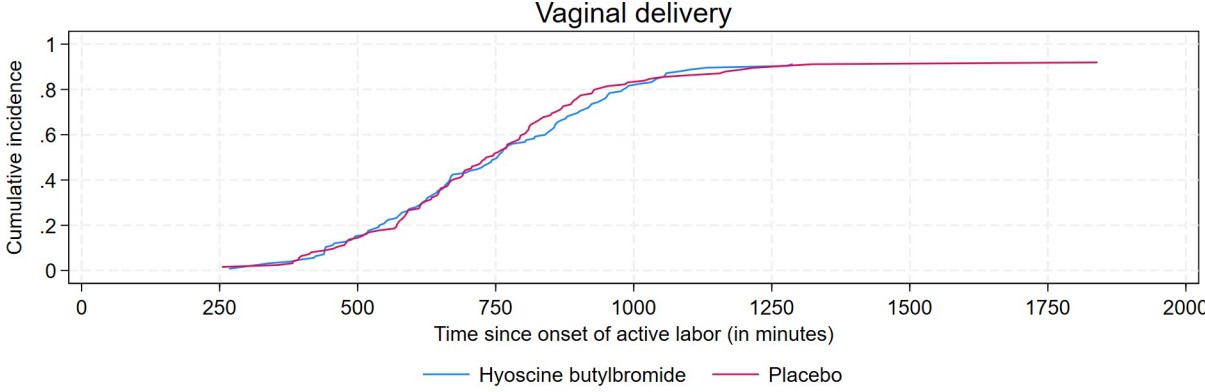

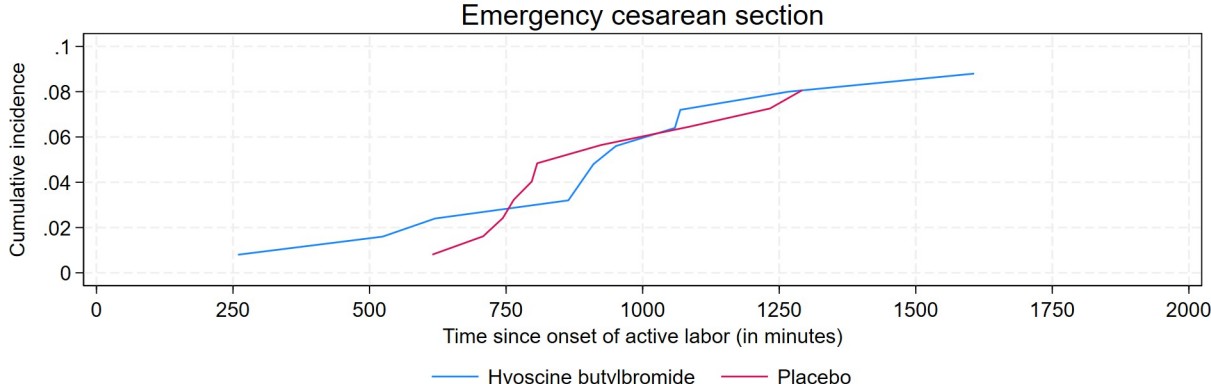

**Fig 6. Cumulative incidences of vaginal delivery and emergency cesarean section as functions of time since onset of active labor for 1 ml (20 mg) hyoscine butylbromide and 1 ml (9 mg) sodium chloride, respectively.** Emergency cesarean section was defined as cesarean section of which an indication developed during labor.

our knowledge, BUSCLAB is the first RCT to assess the effect of hyoscine butylbromide on prevention of slow progress in labor as defined by the WHO partograph. Two meta-analyses of RCTs reported a statistically significant decrease in the duration of labor of almost 1 h for nulliparous women receiving hyoscine butylbromide compared to placebo [29,30]. However, slow labor progress was not an inclusion criterion in these studies. The majority of the included RCTs used hyoscine butylbromide as part of active management of labor, a procedure including artificial rupture of membranes at onset of active labor and augmentation with synthetic oxytocin if progress of labor is slower than 1 cm per hour [37]. This procedure contrasts with the procedure at the study hospital, which follows the WHO recommendations of starting oxytocin infusion if 4 h have passed without progress. Three of the RCTs included in the meta-analysis found no statistically significant effect of hyoscine butylbromide on the duration of labor [38–40]. In the current study, the observed difference in mean duration of labor from IMP administration to any mode of delivery (including emergency cesarean section) between the hyoscine butylbromide group and the placebo group was −5 min. The corresponding result restricted to women with vaginal delivery was −13 min. However, this does not incorporate the inherent incomplete nature of the data when considering duration of labor from IMP administration to vaginal delivery, as the time is not fully observed for women with emergency cesarean section.

In our study, we aimed to include women at risk of prolonged labor by crossing the alert line of the WHO partograph, as these women are have higher occurence of operative delivery

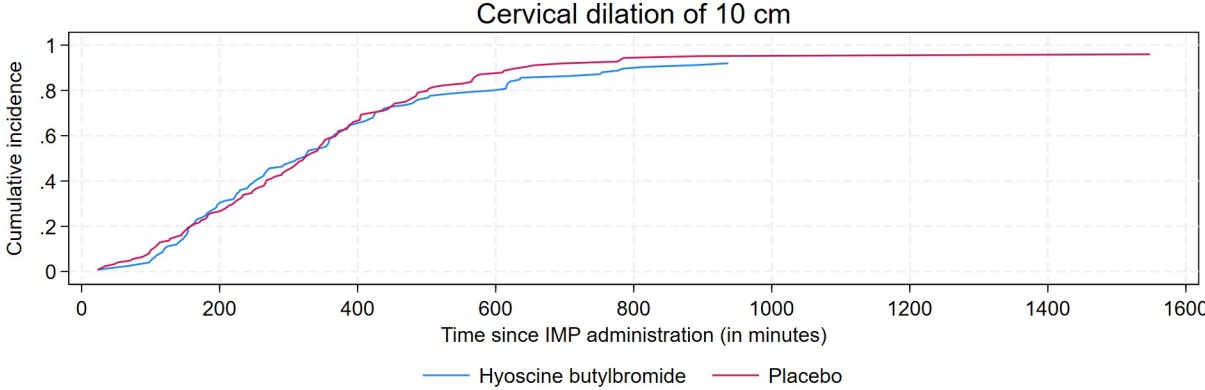

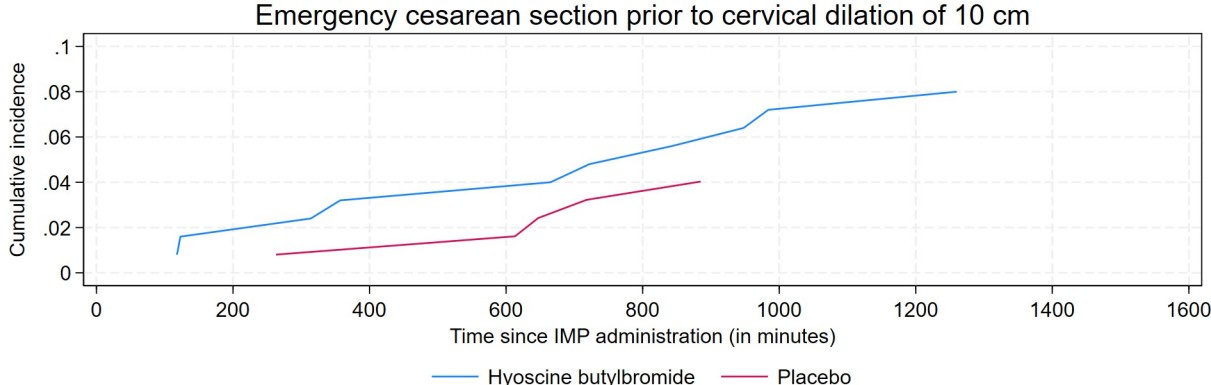

**Fig 7. Cumulative incidences of cervical dilation of 10 cm and emergency cesarean section prior to cervical dilation of 10 cm as functions of time since administration of the IMP for 1 ml (20 mg) hyoscine butylbromide and 1 ml (9 mg) sodium chloride, respectively.** Emergency cesarean section was defined as cesarean section of which an indication developed during labor. IMP, investigational medicinal product.

and prolonged labor. Research by the WHO found that 19.2% of nulliparous women crossed the alert line and 11.7% crossed the action line [41]. Among the women crossing the alert line only, the rate of cesarean section was 5% and 20% had an operative vaginal delivery. Among the women also crossing the action line, the cesarean section rate was 26%, and the vaginal operative rate was 25%. In comparison, for the women not crossing any of the 2 lines, the cesarean section rate was 0.7% and the operative vaginal delivery rate was 11% [41].

Maged and colleagues studied the effect of hyoscine butylbromide on Egyptian women with slow labor [33]. The study found a statistically significant reduction in the mean duration of labor from IMP administration to a fully dilated cervix of 10 cm associated with hyoscine butylbromide when compared to placebo: 322 min versus 451 min [33]. The authors described a stepwise inclusion, where slow labor was defined as a cervical dilation rate of less than 1.2 cm per hour. A group of 187 out of 1,168 women fulfilled this criterion, 56 of whom were delivered by cesarean section. Thereafter, 31 women responded to hydration and oxytocin, leaving 100 women to be included in the study. In our study, there was only a single inclusion criterion regarding slow labor: a cervical dilation rate lower than 1 cm per hour in the active phase of labor. Hence, the study sample in the study by Maged and colleagues may differ from that of our study. In the former study, a single intravenous dose of 40 mg hyoscine butylbromide was used, which is twice as high as in the present study.

The prespecified dose of 20 mg hyoscine butylbromide may have been too low to affect the duration of labor in our sample of first-time mothers already showing signs of slow labor. In

**Table 3. Numbers and percentages of women with adverse events within each system organ class, overall and by treatment group.**

| System organ class | Total (n = 255) | Hyoscine butylbromide (n = 129) | Placebo (n = 126) | p |
|---|---|---|---|---|
| *Immune system disorders* | 3 (1.2%) | 2 (1.6%) | 1 (0.8%) | 1.000[2] |
| Dyspnoea | 1 | 1 | 0 | |
| Pruritus | 1 | 0 | 1 | |
| Chest discomfort | 1 | 1 | 0 | |
| Cough | 1 | 0 | 1 | |
| *Neurological disorders* | 11 (4.3%) | 5 (3.9%) | 6 (4.8%) | 0.728[1] |
| Malaise | 2 | 1 | 1 | |
| Lightheadedness | 2 | 1 | 1 | |
| Headache | 3 | 2 | 1 | |
| Giddiness | 3 | 1 | 2 | |
| Trembling | 1 | 0 | 1 | |
| *Eye disorders* | 9 (3.5%) | 8 (6.2%) | 1 (0.8%) | 0.036[2] |
| Visual disturbance | 9 | 8 | 1 | |
| *Cardiac disorders* | 54 (21.2%) | 51 (39.5%) | 3 (2.4%) | <0.001[1] |
| Maternal tachycardia | 54 | 51 | 3 | |
| *Vascular disorders* | 6 (2.4%) | 2 (1.6%) | 4 (3.2%) | 0.443[2] |
| Hypotension | 2 | 0 | 2 | |
| Hypertension | 1 | 1 | 0 | |
| Dizziness | 2 | 1 | 1 | |
| Flushing | 1 | 0 | 1 | |
| *Gastrointestinal disorders* | 13 (5.1%) | 8 (6.2%) | 5 (4.0%) | 0.418[1] |
| Nausea | 5 | 2 | 3 | |
| Vomiting | 3 | 2 | 1 | |
| Dry mouth | 6 | 5 | 1 | |
| *Renal and urinary disorders* | 139 (54.5%) | 71 (55.0%) | 68 (54.0%) | 0.864[1] |
| Urinary retention | 139 | 71 | 68 | |
| *Other disorders* | 2 (0.8%) | 1 (0.8%) | 1 (0.8%) | 1.000[2] |
| Increased labor pain | 1 | 0 | 1 | |
| Uterine tachysystole | 1 | 1 | 0 | |

[1]Pearson's chi-square test.

[2]Fisher's exact test.

the systematic review by Mohaghegh and colleagues including 3,108 participants [29], the authors performed a sub-analysis comparing hyoscine butylbromide doses of 40 mg and 20 mg with respect to duration of labor. Both doses were shown to shorten the mean duration of the first stage of labor: difference of 69 min with a dose of 40 mg compared to placebo versus 61 min with a dose of 20 mg compared to placebo.

The percentages of women giving birth 2, 4, 6, 8, 10, and 12 h after IMP administration in our study were quite similar in the 2 treatment groups but with a marked difference at 6 h in favor of the hyoscine butylbromide group. This may suggest that hyoscine butylbromide might affect the cervical dilation rate but that the effect is temporary. The half-life of the drug is short (29 min) [42], so increased and/or repeated doses might be necessary to achieve the desired effect on duration of labor.

Our study participants had a mean duration of active labor of 744 min, exceeding the WHO's 12-h definition of prolonged active phase of labor [1]. By comparison, the mean

duration of active labor in the general population of first-time mothers in our hospital was 385 min in 2014 [17]. It is possible that women with prolonged labor as in the present study would have benefited from repeated doses of hyoscine butylbromide. According to the summary of product characteristics, hyoscine butylbromide can be administered in doses of 20 mg and repeated after 30 min with a maximum daily dose of 100 mg [43]. The medication also has a short-lasting effect [42]. Thus, the effect could have been improved if hyoscine butylbromide had been administered repeatedly, and later studies should consider a procedure with repeated doses.

The mean age of the trial participants in our study was higher than in the studies by Mohaghegh and colleagues, Riemma and colleagues, and Maged and colleagues: 32 years versus 26, 25.5, and 24 years, respectively [29,30,33]. These 3 studies included both nulliparous and multiparous women, implying that the mean age of nulliparous women alone must have been even lower. Advanced maternal age of first-time mothers is a well-known risk factor for prolonged labor [44]. Thus, the trial participants in the present study were more prone to prolonged labor than participants in the previous studies. Still, if hyoscine butylbromide had the hypothesized effect, one might anticipate that the effect would be even more pronounced in our study sample of women at higher risk of prolonged labor. On the other hand, it is possible that such a treatment is not appropriate or sufficient for these women, and there might be other reasons for slow progress of labor in this population.

In all of the studies included in the meta-analysis by Riemma and colleagues [30] and the systematic review by Mohaghegh and colleagues [29], IMP was given at onset of active labor (cervical dilation of 3 to 5 cm). This is in contrast to our study, where hyoscine butylbromide was given whenever slow progress of labor was identified and at a cervical dilation of 3 to 9 cm.

Our finding of decreased maternal blood loss in the hyoscine butylbromide group is in accordance with previous research by Imaralu and colleagues, who also found a statistically significant reduction in postpartum hemorrhage associated with hyoscine butylbromide [45]. However, the observed association might be spurious given that postpartum hemorrhage was a secondary outcome in the present study, and the association was also no longer statistically significant in the post hoc adjusted sensitivity analysis. Theoretically, the antispasmodic effect of hyoscine butylbromide may be associated with relaxation of smooth muscle, uterine atony, and postpartum hemorrhage. A short-lasting increase in maternal pulse is one of few side effects of hyoscine butylbromide [43]. A paper on side effects of hyoscine butylbromide based on the BUSCLAB study will be published later.

Our study provides novel information on oxytocin by measuring the exact dose that was given for labor augmentation; 88.0% (110/125) in the hyoscine butylbromide group and 84.7% (105/124) in the placebo group received oxytocin during labor. We also recorded the total time of oxytocin augmentation (in minutes) for each study participant, excluding time periods corresponding to a temporary cessation of augmentation with oxytocin. Our hypothesis was that the hyoscine butylbromide group would require lower doses of oxytocin due to more efficient contractions and a shorter duration of the first stage of labor. However, we found no statistically significant association between treatment group and amount of infused oxytocin, which is in line with our observation that hyoscine butylbromide did not seem to shorten the duration of labor.

An advantage of our study compared to Maged and colleagues is that we used survival analysis (Weibull regression and Cox regression) for all time-to-event outcomes of duration of labor, thereby taking into account the incomplete observation of labors ending with emergency cesarean section. In the presence of such data, this analytical approach is preferable to, e.g., the two-sample $t$ test and the Mann–Whitney $U$ test, as used by Maged and colleagues

[33]. The present study also has some limitations. One potential weakness was the selection of trial participants. More than half of eligible women were not included; 100 did not wish to participate in the study, and 235 were not included upon admission (Fig 2). The FA set excluded a total of 11 women from the initial ITT set of all randomized women (including both eligible and non-eligible women): 10 non-eligible women erroneously randomized to treatment, 1 eligible woman who later withdrew her signed informed consent, and 1 eligible woman who did not receive the IMP due to reasons unrelated to the allocated treatment. Thus, the FA set was regarded as a modified ITT set. In addition, women giving birth at the study hospital were older than nulliparous women in general. Hence, the study participants may have represented a selected group of women with reduced external validity of the study as a result.

Furthermore, this study was not powered to assess the association between treatment group and mode of delivery. This is a clinically important outcome that needs to be investigated in larger studies. Since we designed the present study, the guidelines for labor progress have changed. In the current guidelines from the WHO, active labor is defined as regular contractions and cervical dilation of 5 cm or more [46]. The use of this definition would have led to shorter duration of labor and shorter time from IMP administration to delivery than reported in the present study. Hence, the effect of hyoscine butylbromide might have been more pronounced, and we may have underestimated the effect of hyoscine butylbromide. Larger and/or repeated doses of the study medication might reduce duration of labor when labor is at risk of being prolonged, a hypothesis well suited for further research.

A substantial number of self-reported and observed mild adverse events were noticed, with a higher occurrence of maternal tachycardia (grouped as cardiac disorders) and visual disturbance (grouped as eye disorders) in the hyoscine butylbromide group. These adverse events align with well-established common side effects of hyoscine butylbromide. However, it is of interest that the most common adverse event was urinary retention, a frequently observed condition following prolonged labor involving high frequency of epidural anesthesia and operative vaginal delivery. However, there was no difference in the proportion of urinary retention between the hyoscine butylbromide group and the placebo group, and such events cannot necessarily be directly attributed to the administration of hyoscine butylbromide.

In this double-blind randomized placebo-controlled trial, we found no evidence of shorter duration of labor associated with a single intravenous dose of 20 mg hyoscine butylbromide when compared to placebo in a sample of first-time mothers at risk of prolonged labor. Reassuringly, our study suggests that hyoscine butylbromide can be administered during labor without adverse effects for the offspring and only mild adverse effects for the mother.

## Supporting information

**S1 CONSORT Checklist. Consolidated Standards of Reporting Trials (CONSORT) Checklist.**
(PDF)

**S1 Text. Study protocol BUSCLAB 001, Version No 14, May 26, 2021.**
(PDF)

**S2 Text. Statistical Analysis Plan, Version 1.0, May 19, 2022.**
(PDF)

**S1 Fig. Trial flow diagram.** (a) Investigational medicinal product (IMP). (b) Physical examination fetus included cardiotocography (CTG) with continuous fetal heart rate tracing. (c) Vital signs mother included blood pressure and pulse. Maternal height and body weight was obtained from the pregnancy chart. (d) Pain measurement using visual analogue score. (e)

Physical examination newborn included an examination of general appearance.
(PDF)

**S1 Table. Results of all main analyses, sensitivity analyses, and subgroup analyses.**
(XLSX)

**S2 Table. Results of post hoc adjusted sensitivity analyses.**
(XLSX)

## Acknowledgments

We would like to thank all the women that participated in the study and all the midwives that contributed. We are also grateful to Oslo Metropolitan University for financial support.

## Author Contributions

**Conceptualization:** Lise Christine Gaudernack, Mirjam Lukasse, Trond Melbye Michelsen.

**Data curation:** Lise Christine Gaudernack, Angeline Elisabeth Styve Einarsen, Nina Gunnes.

**Formal analysis:** Lise Christine Gaudernack, Nina Gunnes.

**Methodology:** Lise Christine Gaudernack, Nina Gunnes, Trond Melbye Michelsen.

**Project administration:** Angeline Elisabeth Styve Einarsen, Trond Melbye Michelsen.

**Supervision:** Trond Melbye Michelsen.

**Visualization:** Lise Christine Gaudernack, Angeline Elisabeth Styve Einarsen, Nina Gunnes.

**Writing – original draft:** Lise Christine Gaudernack, Angeline Elisabeth Styve Einarsen, Ingvil Krarup Sørbye, Mirjam Lukasse, Nina Gunnes, Trond Melbye Michelsen.

**Writing – review & editing:** Lise Christine Gaudernack, Angeline Elisabeth Styve Einarsen, Ingvil Krarup Sørbye, Mirjam Lukasse, Nina Gunnes, Trond Melbye Michelsen.

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
