## [Editor Report · Decision Letter 0]

25 May 2023

Dear Dr Gaudernack, 

Thank you for submitting your manuscript entitled "The Effect of Intravenous Butylscopolamine Bromide on Slow Progress in Labor (BUSCLAB): A Double-Blind Randomized Placebo-Controlled Trial" for consideration by PLOS Medicine.

Your manuscript has now been evaluated by the PLOS Medicine editorial staff and I am writing to let you know that we would like to send your submission out for external assessment.

However, we first need you to complete your submission by providing the metadata that are required for full assessment. To this end, please login to Editorial Manager where you will find the paper in the 'Submissions Needing Revisions' folder on your homepage. Please click 'Revise Submission' from the Action Links and complete all additional questions in the submission questionnaire.

Please re-submit your manuscript within two working days, i.e. by May 29 2023 11:59PM.

Once your full submission is complete, your paper will undergo a series of checks in preparation for full assessment

Kind regards,

Richard Turner PhD

Consulting Editor, PLOS Medicine

plosmedicine@plos.org

---

## [Decision Letter · Decision Letter 1]

5 Jul 2023

Dear Dr. Gaudernack,

Thank you very much for submitting your manuscript "The Effect of Intravenous Butylscopolamine Bromide on Slow Progress in Labor (BUSCLAB): A Double-Blind Randomized Placebo-Controlled Trial" (PMEDICINE-D-23-01299R1) for consideration at PLOS Medicine. 

Your paper was evaluated by an asscociate editor and discussed among all the editors here. It was also discussed with an academic editor with relevant expertise, and sent to independent reviewers, including a statistical reviewer. The reviews are appended at the bottom of this email and any accompanying reviewer attachments can be seen via the link below:

[LINK]

In light of these reviews, I am afraid that we will not be able to accept the manuscript for publication in the journal in its current form, but we would like to consider a revised version that addresses the reviewers' and editors' comments. Obviously we cannot make any decision about publication until we have seen the revised manuscript and your response, and we plan to seek re-review by one or more of the reviewers. 

We expect to receive your revised manuscript by Jul 21 2023 11:59PM. Please email us (plosmedicine@plos.org) if you have any questions or concerns.

We look forward to receiving your revised manuscript. 

Sincerely,

Alexandra Schaefer, PhD

PLOS Medicine

plosmedicine.org

GENERAL COMMENTS

Please respond to all editor and reviewer comments.

Please cite the reference numbers in square brackets (e.g., “We used the techniques developed by our colleagues [19] to analyze the data”). Citations should be preceding punctuation.

Please cite your Supporting Information as outlined here: https://journals.plos.org/plosmedicine/s/supporting-information

We suggest referring to Buscopan® (Buscopan) as butylscopolamine bromide throughout your manuscript. In general, we recommend using the international nonproprietary name (rINN), which in your case would seem to be hyoscine butylbromide.

Please remove the statements following the Acknowledgments from the main text. The data should only be included in the corresponding section in the online submission form.

ACADEMIC EDITOR COMMENTS

Like one of the reviewers, I have never heard of anyone using this treatment so some more information on its use would be helpful.

I also think that it would be useful to present the mean and 95% CI of the difference in labour duration between intervention and control (apologies if they did this and I missed it). They give the hazard ratio. However, the meta-analysis expresses the effect in terms of mean difference. It would help in comparing their results with the meta-analysis to have the mean and 95% CI of their difference. If, for example, the 95% CI of the mean difference in the current study included the point estimate of the mean difference from the meta-analysis, you might conclude that the trial doesn't tell you very much.

EDITOR-IN-CHIEF COMMENTS

For the main text, please present the data from the intention-to-treat (ITT) analysis and explain how you accounted for missing follow-up in your analysis. In accordance with the statistical reviewer's comments, please justify the use of Weibull regression (versus Cox proportional analysis) for your survival analysis and demonstrate that Weibull regression is a valid choice for your study. In addition, please provide more details on the clinical background of butylscopolamine bromide in labor (e.g., is it commonly used in labor? Which countries commonly use it?).

COMPETING INTEREST STATEMENT

All authors must declare their relevant competing interests per the PLOS policy, which can be seen here: https://journals.plos.org/plosmedicine/s/competing-interests

For authors with ties to industry, please indicate whether any of the interests has a financial stake in the results of the current study.

DATA AVAILABILITY STATEMENT

PLOS Medicine requires that the de-identified data underlying the specific results in a published article be made available, without restrictions on access, in a public repository or as Supporting Information at the time of article publication, provided it is legal and ethical to do so. Please see the policy at http://journals.plos.org/plosmedicine/s/data-availability and FAQs at http://journals.plos.org/plosmedicine/s/data-availability#loc-faqs-for-data-policy 

The Data Availability Statement (DAS) requires revision. For each data source used in your study: 

ABSTRACT

Please report your abstract according to CONSORT for abstracts, following the PLOS Medicine abstract structure (Background, Methods and Findings, Conclusions); https://www.equator-network.org/reporting-guidelines/consort-abstracts/

Please ensure that all numbers presented in the abstract are present and identical to numbers presented in the main manuscript text.

PLOS Medicine requests that main results are quantified with 95% CIs as well as p values. Please include. When reporting p values please report as p<0.001 and where higher as the exact p value p=0.002, for example. For the purposes of transparent data reporting, if not including the aforementioned please clearly state the reasons why not.

Please include any important dependent variables that are adjusted for in the analyses.

Throughout, suggest reporting statistical information as follows to improve clarity for the reader “22% (95% CI [13%,28%]; p</=)”. Please amend throughout the abstract and main manuscript.

Please note the use of commas to separate upper and lower bounds, as opposed to hyphens as these can be confused with reporting of negative values.

When a p value is given, please specify the statistical test used to determine it. Please report p values as p<0.001 and where higher as 'p=0.002'

Abstract Background:

*Provide the context of why the study is important. The final sentence should clearly state the study question.

Abstract Methods and Findings:

* Please ensure that all numbers presented in the abstract are present and identical to numbers presented in the main manuscript text.

* Please include the study design, population and setting, number of participants, years during which the study took place, length of follow up, and main outcome measures.

* Please include the actual amounts and/or absolute risk(s) of relevant outcomes (including NNT or NNH where appropriate), not just relative risks or correlation coefficients. (example for absolute risks: PMID: 28399126). 

* Please include a summary of adverse events if these were assessed in the study.

Please specify who was blinded to the intervention and control.

Please define the intervention and control states.

Please provide the number in each group.

Please state that analysis was intention to treat.

Please provide the number of participants lost to follow up in each group.

AUTHOR SUMMARY

At this stage, we ask that you include a short, non-technical Author Summary of your research to make findings accessible to a wide audience that includes both scientists and non-scientists. The Author Summary should immediately follow the Abstract in your revised manuscript. This text is subject to editorial change and should be distinct from the scientific abstract. Please see our author guidelines for more information: https://journals.plos.org/plosmedicine/s/revising-your-manuscript#loc-author-summary.

The summary should include 2-3 single sentence, individual bullet points under each of the questions.

It may be helpful to review currently published articles for examples which can be found on our website here https://journals.plos.org/plosmedicine/

INTRODUCTION

ll.84-85: Please reference the studies referred to in the statement.

ll.84-86: Please revise the two sentences to emphasize the aspect of your study investigating the effect of butylscopolamine bromide on slow labor and not labor in general.

Please address past research and explain the need for and potential importance of your study (e.g., more details on the implications of prolonged labor, past research on butylscopolamine bromide in labor). Indicate whether your study is novel and how you determined that. If there has been a systematic review of the evidence related to your study (or you have conducted one), please refer to and reference that review and indicate whether it supports the need for your study.

METHODS AND RESULTS

Please complete the CONSORT checklist and ensure that all components of CONSORT are present in the manuscript, including [how randomization was performed, allocation concealment, blinding of intervention, definition of lost to follow-up, power statement]. Please include the completed CONSORT checklist as Supporting Information.

In accordance with ICMJE requirements, PLOS Medicine requires prospective, public registration of a data sharing plan (as part of mandatory clinical trials registration) for all clinical trials that began enrollment on or after January 1, 2019.

The main analysis should be intention to treat (ie, all individuals randomized are included in the analysis in the groups to which they were originally assigned. If the study included dropouts, specify whether their data are imputed and if so using what method. Please refer to as modified ITT). 

PLOS Medicine requests that main results are quantified with 95% CIs as well as p values. Please include. When reporting p values please report as p<0.001 and where higher as the exact p value p=0.002, for example. For the purposes of transparent data reporting, if not including the aforementioned please clearly state the reasons why not.

Please include any important dependent variables that are adjusted for in the analyses.

Suggest reporting statistical information as detailed above – see under ABSTRACT

Please present numerators and denominators for percentages, at least in the Tables [not necessarily each time they're mentioned].

Please report the number of patients, and account for all methods used in your study.

Please define the length of follow up (eg, in mean, SD, and range).

Please provide the actual numbers of events for the outcomes, not just summary statistics or ORs.

In the trial registry, the secondary outcomes measure ‘changes in maternal and fetal heart rates 30 minutes after IMP administration (in beats/min)” and the secondary dichotomous outcomes are not listed and appear to differ from the primary and secondary outcome measures in the submitted manuscript. Please clarify and explain the discrepancy. If the outcomes were not prespecified in the protocol, please indicate that they were post hoc and explain why they were added. Post hoc comparisons should be presented as hypothesis generating rather than conclusive.

In addition, the trial registration/protocol lists the outcomes of ‘Pain scores using a Visual Analogue Scale at baseline and 30 minutes after administration of IMP’, ‘Urinary retention, defined as need for urinary catheter before the participants leave the delivery ward’, ‘Anal sphincter injury’, ‘Birth experience measured by the validated questionnaire Child Birth Experience Questionnaire’. (a) Can you please present those results as part of this manuscript, or indicate why that is not possible? (b) Can you please indicate when you plan to publish those results?

Please present the safety data for the study including numbers of specific events and whether or not adverse events are thought to be related to treatment.

Please include the study protocol document and analysis plan, with any amendments, as Supporting Information to be published with the manuscript if accepted.

l.211: “[…] admission of the newborn to the NICU within the first hours after birth.” – In the trial registry, this outcome measure is defined as ‘Within 2 hours after birth’, please revise.

ll.332-334: We suggest adding details about the treatment group in which an association for postpartum hemorrhage ≥ 1500 mL in one of the sensitivity analyses was seen. 

DISCUSSION

Please present and organize the Discussion as follows: a short, clear summary of the article's findings; what the study adds to existing research and where and why the results may differ from previous research; strengths and limitations of the study; implications and next steps for research, clinical practice, and/or public policy; one-paragraph conclusion. Please remove any subheadings.

l.414: Please add a period at the end of the sentence.

FIGURES

For all Figures, please ensure that you have complied with our figures requirements http://journals.plos.org/plosmedicine/s/figures.

Please provide titles and legends for all figures (including those in Supporting Information files).

Please consider avoiding the use of red and green in order to make your figure more accessible to those with colour blindness 

In the flow diagram, please indicate the number of individuals in each group analyzed in the ITT analysis. In addition, please change the abbreviation for ‘months’ from ‘mo’ to ‘mos’.

Please in the figure legend/description, define abbreviations used in each figure (including those in Supporting Information files).

Figure 1: The quality of Figure 1 is low. 

Figure 2: Please define ‘NaCl’.

Figure 3: Please add a unit for percentage. 

Figure S5: Figure S5 (Supplementary Material S5) appears not be cited in the main text of your manuscript. Please revise. 

Figure S5: Please define ‘NaCl’. Please change ‘45 unkonown’ to 45 unknown’, ‘1 missing signed consentform’ to 1’ missing signed consent form’, ‘1 not given IMP due to maternal illness not mentioned in exclution criteria’ to ‘‘1 not given IMP due to maternal illness not mentioned in exclusion criteria’, 

TABLES

Please note the use of commas to separate upper and lower bounds, as opposed to hyphens as these can be confused with reporting of negative values. Suggest reporting statistical information as detailed above – see under ABSTRACT

Please provide titles and legends for all tables (including those in Supporting Information files).

Please define all abbreviations used in the table below each table (including those in Supporting Information files).

Table 1: Please remove the p-values presented in Table 1 and the according footnotes.

Table S4: Please define ‘CTG’, ‘IMP, ‘VAS’

REFERENCES

PLOS uses the numbered citation (citation-sequence) method and first six authors, et al.

Please ensure that journal name abbreviations match those found in the National Center for Biotechnology Information (NCBI) databases (http://www.ncbi.nlm.nih.gov/nlmcatalog/journals), and are appropriately formatted and capitalised.

Please also see https://journals.plos.org/plosmedicine/s/submission-guidelines#loc-references for further details on reference formatting. 

Where website addresses are cited, please specify the date of access. 

Comments from the reviewers:

Reviewer #1: This study assess the effect of butylscopolamine bromide on the duration of the active phase of labor in primiparous women showing early signs of slow labor.

Table 1 - Report either mean or median depending on whether you have a gaussian or non-gaussian distribution. This accounts fopr several of your variables.

Table 2 - report either mean or median depending on whether you have a gaussian or non-gaussian distribution.

Table 2 - Why do you report OR and not RR, when reporting results from an RCT?

Table 2 - suggest information on missing values moved to a footnote below the table, to improve readability. 

Table 2 - instead of reporting PPH > 1000 ml yes n (%) or no n(%), simply just report n(%) for the women with pph >1000 ml. This goes for all outcomes. Report the positive findings, so instead of Apgar <7 at 5 minutes - no, please report the actual number and percentage who has a low apgar.

I am not sure why you should adjust for maternal heart rate prior to intervention. Your randomization should have equally distributed your participants into two groups. When looking at your table 1, you have no reason to believe otherwise.

Figure (no number) and figure 2 should be merged into one diagram. In my point of view no need for two almost identical diagrams.

Abstract-Introduction 

Suggest leaving out some of the comments on oxytocin and focus on the possible effect of buscopan

Abstract - result

Only report either mean (SD) or median (IQR) - not both. Why do you report HZ and not RR?

Methods 

I might have missed it, but I could locate any detailed information on recruitment, information of eligible (oral?), when signing the informed consent. I have checked the published protocol as well. However, it seems, you only describe a written information 6 weeks prior to due date. When do they sign the informed consent? Are there any delay between randomization and intervention? Do you have any post randomization dropouts due to a possible delay?

Results

There is no need to include information both the text and, in the consort, flow diagram on recruitment. Suggest that you leave the detailed description to the consort flowchart and shorten the initial text of your result section.

Please consider moving the result on the primary outcome further up in the result section. You write 1.5 page before you report your primary finding.

CEQ - I cannot find any information on the result of the CEQ. And is the CEQ translated and validated in a Norwegian setting? In the protocol you refer to the original Swedish questionnaire.

Reviewer #2: Thank you for the opportunity to read and comment on the paper "The Effect of Intravenous Butylscopolamine Bromide on Slow Progress in Labor (BUSCLAB): A Double-Blind Randomized Placebo-Controlled Trial" by Gaudernack et al. 

The trial aim to compare iv administration of a spasmolytic drug (Butylscopolamine Bromide) with placebo (saline) in nulliparous women with early signs of prolonged labour on the duration on labor from trial medication to delivery. 

I congratulate the authors on this trial; it was well-planned and timely registered, and the paper is overall well written. The rationale to perform a trial in a developed counrty seems good. 

First of all, are spasmolytics really commonly administered drugs during labor (L 76-77)? Could the authors give other references than ref. 22 to support this statement, since I am not sure that this is the case in developed countries (it is certainly not used in the UK or in Denmark). 

A very important point in comparing this trial to other trials would be the study population. I am uncertain if this trial actually aims to prevent or treat prolonged labor as the participants are described to "show early signs" of prolonged labor. Does this population differ considerably from that of the previous studies, as the active labor phase was defined from 3 cm cervical dilation, and the mean cervical dilation at inclusion was 5 cm? Deviating from the alert line at 3 or 4 cm is likely to be a common (?) situation in nullips. Could the authors comment on this aspect and refer to studies using the WHO partograph. 

Please also comment on the use of oxytocin, which was approximately four hours' duration in each group. Do the authors have information about the cervical dilation at time of oxytocin administration? Please state the number of women who had oxytocin in each group as these women are likely to be "true" prolonged labor parturients.

I can tell from the protocol that the primary outcome was the mean duration from IMP to delivery, bus this is not very clear from this manuscript. Please add "mean" to the primary outcome. 

The nomenclature primiparous can certainly be used to describe women carrying their first child/during their first labor; however, using this term might be confusing, as it is also used to describe women with one previous labor. I suggest to use the word nulliparous to describe the women in this trial so that there is no doubt to the reader. 

Introduction; I suggest to markedly reduce the information on possible oxytocin disadvantages as it takes up too much space and is not the focus of this study.

Discussion; This paragraph could overall be much better structured into summary, interpretation of the results (including more general discussion of why this trial is negative as compared to previous trials with significant effect of BB rather than discussing one demographic or outcome at a time), weaknesses (i.e. possible bias in the randomization process with envelopes and using an unblinded person to prepare the medication, precision in including the targeted population including concurrent use of oxytocin) and strengths, generalizability, future research (including moving the discussion about to this place in the discussion as studying multiple administrations is a hypothesis from the trial), and conclusion. 

Minor points

The line "However, the effect of 40 mg vs 20 mg was not compared in the meta-analysis" is unclear as it is stated in the previous lines that the meta-analysis did compare these two doses (L 385). Please revise. 

L 388 I suggest to replace "the same" with "our" hospital. 

L 407-409 The authors should consider if the rationale is the other way around; in a population with a higher risk of prolonged labor (i.e. in this case higher maternal in this study) it could be more difficult to show an effect of the intervention. 

Conclusion: The statement "Larger and/or repeated doses of the medication might reduce duration of labor when labor is prolonged, a hypothesis well suited for further research" (L 450-452) is not a conclusion to be drawn from this data and should be deleted or moved. 

Reviewer #3: This is an interesting RCT on the Effect of Intravenous Butylscopolamine Bromide on Slow Progress in Labor. However, there are a few major issues needing attention.

1. The main analysis (called 'FA') is basically complete case analysis. There is no intention to treat analysis at all in the paper. 'Intention to treat' means 'once randomised always analysed', therefore means analysis of 261 randomised participants which was not possible and not done. The excluded 12 participants may introduce bias to the results as randomisation was disrupted, which is a limitation of the study.

2. Statistical analysis: 

a) Firstly, why using Weibull regression for time to event analysis instead of Cox model? Weibull model requires strong assumption that the change in hazard rate is linear. However we didn't see this anywhere, then why Weibull model?

b) Competing risk analysis. As the emergency cesarean section is a competing risk event, it's not adequate just to censor it. This competing risk issue can be addressed in the competing risk framework using, for example, Fine-Gray model. 

c) The primary analysis needs to be adjusted for important/key baseline variables which seems not done in the paper.

3. Sample size calculation. A short paragraph on sample size with important details is needed in the main paper, rather than just refer to the protocol.

[LINK]

---

## [Decision Letter · Decision Letter 2]

20 Oct 2023

Dear Dr. Gaudernack,

Many thanks for your considered and detailed responses to the editors' and reviewers' comments. I have discussed the paper at length with my colleagues and the academic editor, and it has also been seen again by two of the original reviewers. We appreciate your detailed explanation regarding the intention to treat analysis; however, both reviewers still have some concerns that need to be addressed (including on this point). In addition, the editors require that you add the safety data (ie, adverse events) to this manuscript (per CONSORT).

Based on the reviewers’ and editors’ comments, I invite you to submit another revision of the paper that addresses these comments. Please be sure to address all of the general editorial comments at the end of this letter. 

When submitting your revised paper, please once again include a detailed point-by-point response to the editorial comments. As before, please submit a clean version of the paper as the main article file; a version with changes marked should be uploaded as a marked up manuscript. ***Please note when preparing your response, if your article is accepted, you may have the opportunity to make the peer review history publicly available. The record will include editor decision letters (with reviews) and your responses to reviewer comments. If eligible, we will contact you to opt in or out.***

We ask that you submit this second revision by November 10th. If this date is not feasible, or you have any questions, please email me directly (aschaefer@plos.org). However, please note that I will be on holiday from October 20 through November 3rd; if you need to contact me during that time, please reach out to our Executive Editor, Heather Van Epps (hvanepps@plos.org). 

Sincerely,

Alexandra

Alexandra Schaefer, PhD

PLOS Medicine

aschaefer@plos.org

Major editorial points: 

1. Per CONSORT, it is essential that you report the safety data for the trial, including the number of specific events and whether the adverse events are considered to be related to treatment (along with details about how the AEs were recorded in the Methods section). The adverse events should be presented in a table unless they are very brief, in which case it is sufficient to include them as a paragraph in the main text (with appropriate statistics).

2. Regarding the statisticians point about ITT, we suggest that you consider using ‘modified ITT’ to ensure full transparency regarding the participants who were included in the analysis.

Academic editor comments:

You have calculated the mean and 95% CI of the difference in the duration of labor, but you do not include it because of the effect of not observing the full duration of labor in women who had an intrapartum cesarean section. While the explanation is appreciated, if you feel that this measure is not interpretable, it may be best to remove the reference to the meta-analysis results where this is the outcome. It would be good to calculate the difference in exactly the same way as was done in the trials that made up the meta-analysis, compare your results with the meta-analysis, and then discuss the limitations of this as an outcome. If you prefer not to present your own results with this measure, it remains rather questionable whether it makes sense to discuss the same measure from previous studies.

Reviewer comments: 

Reviewer 1:

Thank you for the opportunity to review the revised document. The paper has without doubt improved since last revision. However, I still suggest further improvement prior to submission. These are however minor.

1. Abstract: The abstract should be further condensed to improve the readability. A suggestion could be to follow the CONSORT guidance for abstracts: Hopewell S, Clarke M, Moher D, Wager E, Middleton P, Altman DG, Schulz KF; CONSORT Group. CONSORT for reporting randomised trials in journal and conference abstracts. Lancet. 2008 Jan 26;371(9609):281-3. doi: 10.1016/S0140-6736(07)61835-2. PMID: 18221781.

2. Methods: Trial design: line 144 -145 suggest delete - or move: initiate the whole section by stating: further detailed information on the design can be found in the published protocol (ref).

3. Result section: You have now deleted all information regarding the participant flow for this section, this is not according to the CONSORT guidance.

4. You need to briefly include information on recruitment, randomization and who received intervention and differences/no differences in baseline characteristics.

5. Line 371 You report both median and mean. You should only report one, depending on Gaussian or non-gaussian distribution.

6. Discussion: PPH difference of 50 ml, and maternal heartrate difference of 3 beat/min, are those differences of clinical relevance?

7. Table 1: For your dichotomous outcome - consider only to report one in the table (not both Yes and No) and the number of missing when relevant. I prefer you report "Yes"

8. Table 2: Since you have reported n in the first row for each column, I suggest that you leave out "number of events" in the first three rows. It improves the readability and I prefer not to use "events" about the included women.

9. For your dichotomous outcome - consider only to report one in the table and the number of missing when relevant.

Reviewer 3 (statistics):

Thanks authors for their great effort to improve the manuscript. The authors have addressed some of my concerns well however there are still a few remaining issues needing to be addressed.

1) Re intention to treat (ITT), I appreciate authors have argued with great effort and respect, however I am still not convinced. I would like to insist that the analysis is simply not an ITT analysis. Therefore, can't use the name of ITT and also need to discuss this in the limitation.

2) Re Weibull regression, although it's fine to use once the assumptions are met, still needs to add one or two sentences to explain why Weibull was chosen over Cox in the main paper as it was not the straightforward first choice in the situation.

3) Re 'The primary analysis needs to be adjusted for important/key baseline variables', it is the best practice these days to do the adjustment even if the variables are perfectly balanced as research shows by this way it will narrow down the 95% CI of the estimates (more precise), therefore the adjustment for main analyses is still recommended.

General editorial points:

1. TITLE: We suggest changing the title to “Evaluation of Intravenous Hyoscine Butylbromide for Slow Progress in Labor (BUSCLAB): A Double-Blind Randomized Placebo-Controlled Trial”

2. ABSTRACT:

a. Please revise your abstract once more be sure to carefully follow the CONSORT Abstract checklist. 

b. Please only report the primary outcomes of the trial in your abstract. We are in the process of overhauling our workflows and policies (with a new Executive Editor), and we now require that trial abstracts only report secondary outcomes if all secondary outcomes of the trial are included. This is to ensure transparent and unbiased reporting and to avoid placing undue emphasis on specific secondary outcomes. For trials that have many secondary outcomes, such as yours, the abstract should be limited to reporting the primary outcome.

c. Please include a brief summary of the adverse events assessed in the study.

d. In the last sentence of the Abstract Methods and Findings section, please describe the main limitation(s) of the study's methodology. 

3. METHODS AND RESULTS: 

a. ll.287-303: This information should be reported in the Results section of the manuscript, rather than the Methods section. 

4. AUTHOR SUMMARY: In the final bullet point of ‘What Do These Findings Mean?’, please describe the main limitations of the study in non-technical language.

5. DISCUSSION:

a. ll. 471-472: For a better understanding, please make it clear that 69 min and 61 min refers to the difference in the first stage of labor compared with placebo (eg, reduced by 69 min…)

b. Causal language - In trials, there is usually a distinction in the language in terms of causal vs associational for primary and secondary trial outcomes, respectively. It would be preferable to use associational language in the discussion( and other sections) when referring to secondary outcomes (e.g., l.520).

c. Please remove the ‘Conclusion’ subheading from the discussion, as the one-paragraph conclusion should be part of the discussion section.

6. Figure 4: Are the lines for the hazard rates (vaginal delivery) of hyoscine butylbromide and placebo congruent (only one line is visible)?

7. Table 1: In the second row (Maternal age (in years)), please either include the unit in parentheses (“years”) and then remove "years" from each age group in the list below; OR remove the unit from the heading and keep "years" as the unit with each age group.

8. Table 1: Please note that the footnote ‘6’ is not formatted as superscript.

9. SUPPLEMENT, S2 Trial Flow Diagram: I cannot locate footnote ‘e’ in the figure description?

10. REFERENCES: Where website addresses are cited, when specifying the date of access please replace the word ‘cited’ with ‘accessed’.

11. Please add the following statement, or similar, to the Methods: "This study is reported as per the Consolidated Standards of Reporting Trial (CONSORT) guideline (S1 Checklist)."

12. Please remove the statements following the Acknowledgments from the main text (Declaration of interest, Role of the funding source, Data sharing). The data should only be included in the corresponding section in the online submission form.

13. Please include this information “Contact email to the study sponsor: mirnyb@ous-hf.no” in the Data availability statement in the manuscript submission form.

14. If you have not already done so, please upload any figures associated with your paper as individual TIF or EPS files with 300dpi resolution at resubmission; please read our figure guidelines for more information on our requirements: http://journals.plos.org/plosmedicine/s/figures. While revising your submission, please upload your figure files to the PACE digital diagnostic tool, https://pacev2.apexcovantage.com/. PACE helps ensure that figures meet PLOS requirements. To use PACE, you must first register as a user. Then, login and navigate to the UPLOAD tab, where you will find detailed instructions on how to use the tool. If you encounter any issues or have any questions when using PACE, please email us at PLOSMedicine@plos.org.

---

## [Decision Letter · Decision Letter 3]

22 Dec 2023

Dear Dr. Gaudernack,

Thank you very much for re-submitting your manuscript "The Effect of Intravenous Hyoscine Butylbromide on Slow Progress in Labor (BUSCLAB): A Double-Blind Randomized Placebo-Controlled Trial" (PMEDICINE-D-23-01299R3) for review by PLOS Medicine.

Thank you for your detailed response to the editors' and reviewers' comments. I have discussed the paper with my colleagues and the academic editor, and it has also been seen again by the statistical reviewer. The changes made to the paper were satisfactory to the reviewer. As such, we intend to accept the paper for publication, pending your attention to the editorial comments below in a further revision. When submitting your revised paper, please once again include a detailed point-by-point response to the editorial comments.

[LINK]

We expect to receive your revised manuscript within 1 week. Due to the upcoming holiday season, we are happy to provide additional time to complete the revision. Please email me (aschaefer@plos.org) or the journal staff on plosmedicine@plos.org if you have any questions or concerns. 

We look forward to receiving the revised manuscript by Dec 29 2023 11:59PM.   

Sincerely,

Alexandra Schaefer, PhD

Associate Editor 

PLOS Medicine

plosmedicine.org

Requests from Editors:

EDITORIAL POINTS

As discussed, please report the full safety data for the trial, including the number of specific events and whether the adverse events are considered to be related to treatment (along with details about how the AEs were recorded in the Methods section). The adverse events should be presented in a table unless they are very brief, in which case it is sufficient to include them as a paragraph in the main text (with appropriate statistics).

DATA AVAILABILITY

Thank you for providing the Data Availability Statement (DAS) statement. In line with PLOS’ policy on data availability, we kindly ask you to share a “minimal data set”. PLOS defines the “minimal data set” to consist of the data set used to reach the conclusions drawn in the manuscript with related metadata and methods, and any additional data required to replicate the reported study findings in their entirety. Authors do not need to submit their entire data set, or the raw data collected during an investigation. If possible, please submit the following data:

The values behind the means, standard deviations and other measures reported;

The values used to build graphs.

ABSTRACT

1) ll.55-56 “Women giving birth at the study hospital were older and had higher educational level than nulliparous women in general.” – This detail may be more appropriate for inclusion in the Results/Discussion section rather than the Abstract. If stated in the Results/Discussion section, pleas provide reference.

2) l. 57 "...in the study of various reasons." - Please specify "various reasons". Also, according to the flow chart, more than 80% of eligible women were excluded. Editorial suggestion: "More than 80% of eligible women were excluded from the study because they did not meet the inclusion criteria." Please make it clear that this is the main limitation(s) of the study's methodology; the phrase "The main limitation of the study..." may be useful.

AUTHOR SUMMARY

1) Under ‘What Did the Researchers Do and Find?’, please split the two bullet points into four to make the details provided more accessible. Editorial suggestion:

What Did the Researchers Do and Find?

• We performed a double-blind randomized placebo-controlled study, including 249 nulliparous women showing first signs of slow labor progress. 

• The participants were randomized to receive either a single intravenous dose of the spasmolytic drug hyoscine butylbromide (Buscopan®) (20 mg) or a single intravenous dose of saline solution (placebo).

• We found no statistically significant difference in labor duration between the two treatment groups. There was a decrease in postpartum hemorrhage and a slight increase in maternal heart rate in the hyoscine butylbromide group. 

• No maternal serious adverse events were observed, nor did we observe any neonatal adverse events.

2) Under ‘What Do These Findings Mean?’, the first bullet point seems to repeat an observation rather than being an interpretation of the finding. Editorial suggestion: Hyoscine butylbromide was not found to impact duration of labor from treatment administration to delivery for first-time mothers with long labors.

3) The last bullet under ‘What Do These Findings Mean?’ point should only describe the main limitation of the study's methodology. The sentence starting “Randomized controlled trials assessing the effect of higher or…” may be added as a separate bullet point prior to the limitation.

METHODS AND RESULTS

1) At the beginning of the Results section, please include a few sentences about the study population, i.e., the number screened for eligibility, the number deemed eligible, the number not enrolled, etc. (according to the flowchart).

2) ll. 375-377 “…was markedly higher..”: Did you test for significance? If not, please temper language.

3) l.408: Please use causal language for the description of primary outcomes. In trials, there is usually a distinction in the language in terms of causal vs associational for primary and secondary trial outcomes. It would be beneficial to use associational language in the discussion and other sections for secondary outcomes.

DISCUSSION

1) ll.446-449: Please use causal language for the description of primary outcomes. 

2) ll.474-481: Please provide reference for the first and last sentence of the paragraph if referring to the same reference.

3) l.558: Please specify ‘various reasons’.

4) l.564: “In addition, women giving birth at the study hospital were older and had higher educational level than nulliparous women in general.” - Please provide reference.

REFERENCES

Please thoroughly revise all references and ensure that journal name abbreviations match those found in the National Center for Biotechnology Information (NCBI) databases (http://www.ncbi.nlm.nih.gov/nlmcatalog/journals), and are appropriately formatted and capitalised (e.g., for reference [17] BMC Pregnancy and Childbirth should be BMC Pregnancy Childbirth)

FIGURES

1) Figure 2: Please remove any CONSORT labeling. The top of the flowchart should show the enrollment/assessment of eligibility box.

2) Figure 3: Please remove the headings ‘Absolute frequency’ and ‘Relative frequency’.

3) Figure 4: Please ensure to present 95% confidence intervals in the figure. 

4) Figure 4: Please note that the lower graph of Figure 4 is titled ‘emergency cesarean section delivery’ while in other figures you use the title ‘emergency cesarean section’. Please revise.

5) Figure 3-7: Please in the figure description, add details about the dose of hyoscine butylbromide, the definition of placebo as well as the definition of ‘emergency cesarean section’.

SOCIAL MEDIA

To help us extend the reach of your research, please provide any X (formerly known as Twitter) handle(s) that would be appropriate to tag, including your own, your coauthors’, your institution, funder, or lab. Please respond to this email with any handles you wish to be included when we tweet this paper.

Comments from Reviewers:

Reviewer #3: Thanks authors for their great effort to improve the manuscript. I am satisfied with the response and revision. No further issues needing attention.

[LINK]

---

## [Editor Report · Decision Letter 4]

25 Jan 2024

Dear Dr Gaudernack, 

On behalf of my colleagues and the Academic Editor, Gordon C Smith, I am pleased to inform you that we have agreed to publish your manuscript "The Effect of Intravenous Hyoscine Butylbromide on Slow Progress in Labor (BUSCLAB): A Double-Blind Randomized Placebo-Controlled Trial" (PMEDICINE-D-23-01299R4) in PLOS Medicine.

I appreciate your thorough responses to the reviewers' and editors' comments throughout the editorial process. We look forward to publishing your manuscript, and editorially there are only a few remaining stylistic/presentation points that should be addressed prior to publication. We will carefully check whether the changes have been made. If you have any questions or concerns regarding these final requests, please feel free to contact me at aschaefer@plos.org.

Please see below the minor points that we request you respond to:

1) ll.452-459: Thank you for providing the full safety data. In the main text, we ask you to include one or two sentences detailing the most common adverse events (e.g., those affecting >10% of the study population).

2) Table 3: We feel that the adverse events should be presented in a more conventional manner. Please state the specific disorders/conditions/AEs included in each of the organ-specific categories. You do not need to include a "No" category. For guidance, we recommend Table 2 of the paper by Bruinsma A et al. (2022; doi: 10.1016/j.eurox.2022.100165).

2) Figure 2: Please change "1 mL 20 mg hyoscine butylbromide" to "1 mL (20 mg/mL) hyoscine butylbromide" and "1 mL sodium chloride" to "1 mL (9 mg/mL) sodium chloride".

3) Figure 4: In the figure description, we suggest adding a note that the lines for vaginal delivery and emergency cesarean section are congruent.

4) The supplementary files "S2 Experimental Flow Diagram" and "S5 Results" were not submitted. Please be sure to submit all files.

5) Please be sure to update the Data Availability and Competing Interest fields in the online submission form according to the files you provided as supporting information.

PRESS

Sincerely, 

Alexandra Schaefer, PhD 

Associate Editor 

PLOS Medicine